# Long-Term Administration of Antioxidant N-Acetyl-L-Cysteine Impacts Beta Cell Oxidative Stress, Insulin Secretion, and Intracellular Signaling Pathways in Aging Mice

**DOI:** 10.3390/antiox14040417

**Published:** 2025-03-31

**Authors:** Meg Schuurman, Jonathan Nguyen, Rachel B. Wilson, Malina Barillaro, Madison Wallace, Nica Borradaile, Rennian Wang

**Affiliations:** 1Children’s Health Research Institute, London, ON N6C 2V5, Canada; mschuur2@uwo.ca (M.S.); rwilso89@uwo.ca (R.B.W.); mbarilla@uwo.ca (M.B.); mwalla32@uwo.ca (M.W.); 2Department of Physiology & Pharmacology, University of Western Ontario, London, ON N6A 3K7, Canada; jnguy253@uwo.ca (J.N.); nica.borradaile@schulich.uwo.ca (N.B.)

**Keywords:** N-acetyl-L-cysteine (NAC), beta cell oxidative stress, exocytosis proteins, pancreatic stellate cells (PaSCs), collagen fiber, HFD-stress challenge

## Abstract

Research into the effects of long-term antioxidant supplementation on the islet microenvironment is limited. This study examined whether long-term N-acetyl-L-cysteine (NAC) supplementation can prevent changes in metabolic outcomes, beta cell function, and pancreatic stellate cell (PaSC) activation in aging mice. Male C57BL/6N mice at 18 weeks were administered 50 mM NAC through their daily drinking water and treated for up to 60 weeks. Aging NAC mice displayed lower body weights and improved glucose tolerance but reduced insulin secretion and insulin signaling compared to control (ND) mice. When some 40-week-old ND and NAC mice were subjected to 8 weeks of a high-fat diet (HFD)-stress challenge, results showed that NAC reduced HFD-induced beta cell oxidative stress and preserved nuclear PDX-1 expression. The findings from this study suggest that while NAC can be beneficial for diet-induced stress during aging, the effects of long-term NAC on the islets of physiologically aging mice are more ambiguous. Further exploration is required to determine the effects of NAC-mediated lowering of beta cell oxidative stress on insulin secretion and signaling pathways. This study highlights the importance of investigating oxidative stress balance in aging islets under normal diet conditions to determine if antioxidative therapies can be utilized without interfering with essential physiological processes.

## 1. Introduction

Aging has been determined to be a significant risk factor for type 2 diabetes mellitus (T2D), independent of lifestyle and obesity. The risk for T2D development begins to exponentially rise beyond age 65 [1]. However, aging and elderly individuals are often excluded from clinical studies due to upper age limits or multiple comorbidities [2]. In humans, there is a positive correlation between glucose intolerance and age [3]. It is well established that the natural aging process is associated with elevated oxidative stress levels, which is a major contributing factor to T2D development [4]. Aging beta cells are at increased risk for the development of abnormal function, including reduced glucose-stimulated insulin secretion (GSIS) and beta cell hypertrophy, and are susceptible to increased apoptosis [5,6]. Studies show that an aging pancreas has elevated levels of mitochondrial oxidative stress and of alpha-smooth muscle actin (αSMA) expression, a sign of pancreatic stellate cell (PaSC) activation. Activated PaSCs are associated with the secretion of inflammatory cytokines and excessive ECM production, leading to pancreatic inflammation and fibrosis [7,8], and people with T2D have been shown to have increased activated PaSCs compared to heathy controls [9].

In light of the relationship between oxidative stress and T2D development, oxidative stress is often examined in the context of islet biology. Oxidative stress occurs due to the production of reactive oxygen species (ROS), including superoxide radicals, hydrogen peroxide, and hydroxyl radicals, which exceeds cellular antioxidant capacity. ROS are thought to be produced in beta cells from the mitochondrial electron transport chain upon exposure to both high and low glucose [10]. Although excess levels of ROS can promote beta cell dysfunction, a certain amount of ROS is necessary to support beta cell functions. Compared to other organs, the pancreas has increased susceptibility to oxidative stress due to lower antioxidant enzyme expression [11], and beta cells are more susceptible to oxidative stress damage than other islet cell types [12]. Furthermore, T2D pancreata were shown to have reduced superoxide dismutase (SOD) expression, augmenting susceptibility to oxidative stress as demonstrated by the presence of 8-OHdG, a marker of oxidative stress-induced DNA damage [12]. Overall, the limited antioxidant capacity of beta cells combined with the increased presence of oxidative stress in aging and states of overnutrition likely contributes to the beta cell dysfunction observed in T2D.

Antioxidant therapy is often studied in T2D research due to rising levels of oxidative stress and reduced antioxidant capacity in people living with T2D [13]. Specifically, people with T2D have reduced glutathione (GSH) compared to unaffected individuals [14]. N-acetyl-L-Cysteine (NAC) is a thiol-containing compound and a precursor to GSH, which provides its direct and indirect antioxidant effects, respectively [13]. NAC has been shown to increase serum GSH levels and significantly reduce serum markers of oxidative stress [15]. Our previous study determined that NAC improved glucose tolerance and insulin sensitivity in high-fat diet (HFD)-induced diabetic mice in a time- and dose-dependent fashion, which was associated with a reduction in beta cell oxidative stress, intra-islet pancreatic stellate cell (PaSC) activation and islet fibrosis [16]. However, there is a lack of investigation into the effects of long-term NAC on pancreatic beta cells and intra-islet PaSCs in the context of physiological aging.

In the present study, we aimed to investigate the effects of long-term NAC administration on metabolic parameters and islet biology in the context of aging under normal diet conditions and how mice receiving long-term NAC treatment cope with HFD-stress challenge. We hypothesized that long-term oral administration of NAC to mice would maintain metabolic outcomes and beta cell function by reducing beta cell oxidative stress and intra-islet PaSCs activation during aging and with a diet-stress challenge.

## 2. Materials and Methods

### 2.1. Mouse Model of Aging Study with NAC Administration

Male C57BL/6N (B6N) mice purchased from Charles River (Charles River Laboratories, Montreal, QC, Canada) were maintained on a normal chow diet composed of 22% kcal from fat, 23% kcal from protein, and 55% kcal from carbohydrates (Harlan Teklad, Indianapolis, IN, USA). At 18 weeks, two experimental groups were generated (Appendix A): (1) control mice maintained on a normal diet (ND); and (2) mice treated with the antioxidant NAC on a normal diet (NAC). NAC (Santa Cruz Biotechnology, Inc., Santa Cruz, CA, USA) was administered in drinking water at a concentration of 50 mM. All n-values represent litter-matched control and NAC treated mice from multiple in vivo experiments. The concentration of NAC was determined from our previous study, which investigated the use of both unmodified 10 mM and 50 mM NAC in drinking water [16]. Additionally, 50 mM NAC in drinking water resulted in a daily intake of NAC in the range of ~600–1200 mg/kg, which was determined in other research to improve metabolic outcomes in mice [17]. Both ND and NAC mice were analyzed at 28, 36, 48, or 60 weeks, and some analyzed data of ND mice at 28 and 36 weeks were previously reported [16].

At 40 weeks, a group of ND and NAC mice received a HFD challenge for an 8-week period to form ND^HFD^ and NAC^HFD^ experimental groups (Appendix A). HFD is composed of 60% kcal from fat, 20% from protein, and 20% from carbohydrates (Research Diets INC, New Brunswick, NJ, USA).

NAC intake was assessed in each study timepoint over a three-day period by measuring water consumption (Appendix A) [16]. Fat pad mass was measured in mice at 60 weeks. Weekly body weight and health checks were performed on all mice, and any that developed health concerns were excluded from the study. All animal work was conducted based on approved protocols from the University of Western Ontario Animal User Subcommittee in accordance with the Canadian Council of Animal Care guidelines.

### 2.2. Metabolic Studies in Experimental Mouse Models

At the last week of the assigned study period, 4 h (h) fasting blood glucose was examined and followed by intraperitoneal glucose or insulin tolerance tests (IPGTT or IPITT), or glucose-stimulated insulin secretion (GSIS) tests as previously described [16].

To determine in vivo GSIS, blood samples were collected following a 4 h fast (0 min) and at either 5 and 35 min or 10 and 30 min after intraperitoneal injection of glucose (2 mg/g body weight). For ex vivo GSIS, thirty isolated islets from ND and NAC mouse pancreata at 60 weeks were handpicked, size-matched (10 small, 10 medium, 10 large), and recovered overnight in RPMI-1640 medium containing 11 mM glucose plus 10% FBS. Islets were sequentially treated for two successive incubations of 1 h in low (2.2 mM), followed by 1 h in high glucose (22 mM) in RPMI-1640 media with 0.5% BSA [18]. Plasma insulin levels from fed or fasting cardiac blood, GSIS, and islet insulin content were measured using STELLUX^®^ Chemi Rodent Insulin ELISA (ALPCO, Salem, NH, USA) [16]. Total insulin content was normalized the total islet protein content (μg/mg). Plasma triglycerides and cholesterol were measured using enzymatic colorimetric assays (Wako Diagnostics, Mountain View, CA, USA) [16].

### 2.3. Immunohistological Staining and Morphometric Analyses

At the end of each experimental timepoint, pancreata from mice were dissected and processed for paraffin embedding and immunostaining as previously described [19]. Sections were incubated with the appropriate dilutions of primary antibodies, as listed in Appendix A. Secondary antibodies were conjugated to fluorescein isothiocyanate (FITC) and tetramethyl rhodamine isothiocyanate (TRITC) (1:50; Jackson Immunoresearch, West Grove, PA, USA) for immunofluorescence detection. 4′-6′-diamidino-2-phenylindole (DAPI, Sigma) was used for nuclear counterstaining. Stained sections were imaged and analyzed for islet density and alpha and beta cell mass as previously described [16,20]. Representative immunofluorescence images were captured using a Nikon Eclipse Ti2 Confocal Microscope (Nikon, Melville, NY, USA). PDX-1, 8OHdG, GLUT2, and insulin exocytosis proteins were identified by double immunofluorescence staining with insulin+ cells and quantified from at least 10 random islets per pancreatic section. To quantify the percentage of the activated intra-islet PaSCs population, labeled by αSMA, the positive intra-islet αSMA^+^ cell area (µm^2^) was manually traced using Fiji and divided by the total insulin-positive area (µm^2^) [16].

A trichrome staining kit (Abcam, Cambridge, MA, USA) was used to measure intra-islet collagen deposition [16]. Stained images were acquired using an Aperio AT2 whole slide scanner (Leica Biosystems Inc., Concord, ON, Canada). The percentage of islet collagen deposition was determined using an ImageJ (v1.53t) macro, which segments blue areas (collagen) from red areas in the image using color deconvolution and measures total blue area per image as previously described [16,21].

### 2.4. Protein Quantification and Western Blot Analysis

Islet isolation was conducted at 60 weeks. In brief, 3 mL collagenase V (1 mg/mL, Sigma) was slowly injected into the common bile duct after occlusion of the distal end just proximal to the duodenum. The distended pancreas was excised, and the digestion was performed in a water bath at 37 °C for 30 min. The digestion was stopped by ice-cold Hanks’ balanced salt solution containing 10% FBS and washed thoroughly to remove residual collagenase. Freshly isolated islets from the pancreas of ND and NAC mice at 60 weeks were handpicked under a dissecting microscope and lysed for Western blotting and islet insulin content [22]. Protein was loaded on a 4–20% Mini-PROTEAN TGX Gel (Bio-Rad Laboratories, Hercules, CA, USA) at 20 μg and transferred to a PVDF membrane (Thermo Scientific, Waltham, MA, USA). Membranes were incubated with primary antibodies (Appendix A) and corresponding horseradish peroxidase-conjugated secondary antibodies, and chemiluminescence was induced using Western Lightning^®^ ECL Pro (PerkinElmer Inc., Waltham, MA, USA). The iBright1500 imaging system (Invitrogen, Waltham, MA, USA) was used to image membranes, and iBright Analysis Software (v5.3.0 Invitrogen) was used to conduct densitometry analysis. Data were normalized to total membrane protein or total antibody where applicable [23].

### 2.5. Statistical Analysis

All data are expressed as means ± SEM. Statistical significance was analyzed using GraphPad Prism (version 9 GraphPad Software, San Diego, CA, USA) with a 95% of confidence interval. The difference was analyzed using two-tailed unpaired *t*-tests between each timepoint. Differences were considered to be statistically significant when *p* < 0.05.

## 3. Results

### 3.1. Long-Term NAC Treatment Improves Glucose Tolerance in Aging Mice

Mice administered NAC showed significantly lower body weight throughout the study (Figure 1A,B), and no difference in food intake was detected between ND and NAC mice (Figure 1C). Additionally, NAC mice showed significantly lower fat pad weight at 60 weeks (Figure 1D). Fasting blood glucose at 4 h (Figure 1E) was unchanged between groups, with the exception of the 48-week NAC cohort, and no difference was detected for overnight fasting blood glucose between ND and NAC mice at 60 weeks (Figure 1F). Fed plasma insulin levels were similar between ND and NAC mice at each timepoint (Figure 1G), but significantly lower 4 h fasting plasma insulin was determined in 60-week NAC mice compared to ND mice (Figure 1H). Fed circulating triglycerides and cholesterol showed no differences between ND and NAC mice (Appendix A).

At 28 weeks, both ND and NAC mice showed a similar glucose clearance response when challenged with an IPGTT (Figure 2A). However, 60-week NAC mice normalized their glucose within 2 h, which did not occur in ND mice (Figure 2B). The GTT AUC in NAC mice was maintained throughout the duration of the study, while ND mice showed a progressive impairment at 48 and 60 weeks (Figure 2E). There were no differences in ITT AUC between ND and NAC mice during the study period (Figure 2B,F). In vivo GSIS revealed that both NAC and ND mice at 60 weeks had limited GSIS response (Figure 2G). Notably, NAC mice displayed significantly reduced insulin secretion pre- and post-glucose injection (Figure 2G). Ex vivo islet GSIS further verified that NAC and ND mouse islets showed an impaired response to GSIS (Figure 2H). The insulin content of NAC mouse islets was significantly lower when compared to ND islets at 60 weeks (Figure 2I), which was correlated with lower plasma insulin and insulin release during GSIS (Figure 1H and Figure 2G,H).

### 3.2. Long-Term NAC Treatment Reduces Beta Cell Oxidative Stress and Activated Intra-Islet PaSCs in Aging Mice

To examine NAC’s impact on beta cell oxidative stress, the presence of 8OHdG in beta cells was quantified using double staining for 8OHdG and insulin (Figure 3A). 8OHdG labeling in insulin-positive cells was significantly lower in NAC mice at 28 and 60 weeks compared to ND mice (Figure 3B). The reduction in intra-islet oxidative stress coincided with a significant decrease of intra-islet αSMA^+^ area in 60-week NAC mice (Figure 3C,D), suggesting that NAC lowered intra-islet PaSC activation. To evaluate whether the NAC-associated decrease in PaSC activation resulted in a decrease in islet fibrosis, quantification of islet collagen was performed using a trichrome stain (Figure 3E). Islet collagen deposition was significantly lower in NAC mice compared to ND mice (Figure 3F).

### 3.3. Long-Term NAC Treatment Preserves Beta Cell Mass but Lowers Nuclear PDX-1 Expression in Aging Mice

Double immunofluorescence staining for insulin and glucagon were used for islet morphometric analysis. A similar islet density was observed in ND and NAC mice during the study period (Figure 4A). No differences in islet size distribution were observed between NAC and ND mice at 60 weeks (Figure 4B). However, beta cell mass was increased in 60-week ND mice compared to NAC mice, and beta cell mass in 60-week NAC mice was relatively similar to that of 48-week NAC mice (Figure 4C). There were no discernable changes in alpha cell mass (Figure 4D). When analyzing the transcription factor PDX-1, which is involved in regulating beta cell function and insulin secretion, it was found that nuclear PDX-1 localization was significantly lower in 28- and 60-week NAC mice (Figure 4E,F). The reduced nuclear PDX-1 appears to be correlated with 8OHdG, which was also significantly reduced in NAC mice at 28 and 60 weeks (Figure 3A,B). Western blot analysis of isolated islets for the cell proliferation marker PCNA and cell death markers PARP and caspase 3 showed no detectable cleaved PARP and caspase-3, and no changes in levels of total PCNA, PARP, and caspase-3 between ND and NAC islets at 60 weeks (Appendix A).

### 3.4. Long-Term NAC Treatment Alters Beta Cell Insulin Exocytosis Protein Levels and Intracellular Signaling Pathways in Aging Mice

Due to the observation of lower nuclear PDX-1, beta cell insulin release and insulin content in NAC mice at 60 weeks, the glucose sensor GLUT2 and insulin exocytotic proteins syntaxin 1A, Munc18-1A, SNAP25, and VAMP2 were examined. There were no differences in GLUT2 expression between 60-week ND and NAC islets (Appendix A). However, a significant reduction in the percentage of SNAP25+ area in the islets (Appendix A) was verified by Western blot analysis in NAC islets at 60 weeks compared to ND islets (Figure 5A,B). The expression of exocytotic proteins VAMP2, syntaxin 1A, and its associated chaperone protein Munc18-1A were also significantly reduced in NAC islets at 60 weeks (Figure 5A,B).

Factors involved in beta cell insulin signaling pathways were also examined in 60-week islets. We identified a significant reduction in phosphorylated insulin receptor (p-IRβ^Tyr1146^) and insulin receptor substrate 1 (p-IRS1^Ser612^) in NAC mouse islets, which is linked to a significant reduction in phosphorylated AKT^Ser473^ and ERK1/2^Thr202^/Tyr^204^ (Figure 5C,D). Notably, NAC mouse islets had significantly reduced phosphorylated eIF2α^Ser51^ and NFκB, indicating reduced integrated stress response and inflammatory signaling, respectively, under long-term NAC treatment (Figure 5E,F). These data indicate that a reduced beta cell intracellular ROS level, as seen in NAC islets (Figure 3A,B), could impact insulin exocytotic protein expression and associated signaling pathways. Furthermore, we investigated various proteins involved in inflammatory pathways in isolated islets and found there were no significant differences in PPARγ [24,25,26], TNFα, and Smad2/3 protein levels (Appendix A).

### 3.5. NAC Preserves Glucose Tolerance in Aging Mice Undergoing HFD Challenge

To examine whether long-term NAC administration could protect aging islets facing an HFD-stress challenge, both ND and NAC mice at 40 weeks were fed with an HFD for 8 weeks. Body weights were significantly lower in NAC^HFD^ mice compared to ND^HFD^ mice (Figure 6A,B). There were no significant changes in food intake (Figure 6C) and fasting blood glucose between the two cohorts (Figure 6D). Interestingly, the hyperinsulinemia observed in ND^HFD^ mice was significantly improved in NAC^HFD^ mice (Figure 6E). Indeed, NAC administration during the HFD challenge allowed for maintenance of plasma insulin levels similar to that of non-HFD-fed mice (Figure 1G and Figure 6E). Fed circulating triglycerides and cholesterol were unchanged between ND^HFD^ and NAC^HFD^ mice (Appendix A). In vivo GSIS revealed that both NAC^HFD^ and ND^HFD^ mice showed similar poor responses to GSIS at 10 and 30 min (Figure 6F). However, similar to the findings in normal diet conditions (Figure 2G), NAC administration significantly lowered insulin release over time during GSIS compared to ND^HFD^ mice (Figure 6F). A significant impairment of glucose tolerance was present in ND^HFD^ mice that was not observed in NAC^HFD^ mice (Figure 6G). There was no difference in insulin sensitivity during the HFD challenge between ND^HFD^ and NAC^HFD^ mice (Figure 6H).

### 3.6. NAC Preserved Beta Cell Mass, Nuclear PDX-1, and Reduced Oxidative Stress in Aging Mice Undergoing HFD Challenge

Morphometric analysis of islet density and alpha and beta cell mass showed no significant differences between experimental groups during the HFD challenge (Figure 7A). Notably, GLUT2 staining was localized to the plasma membrane in NAC^HFD^ beta cells, while the staining was more diffuse in ND^HFD^ beta cells (Figure 7B). Similarly, strong syntaxin 1A membrane localization was observed in NAC^HFD^ beta cells compared to ND^HFD^ beta cells (Figure 7C). These data suggest that, under the HFD challenge, NAC preserved membrane localization of glucose-sensing and insulin exocytosis machinery, perhaps by mitigating HFD-induced oxidative stress. The HFD challenge induced beta cell dysfunction as shown by an apparent loss of nuclear PDX-1 signal in ND^HFD^ mice (Figure 7D) compared to ND mice with no HFD challenge (82% in ND^HFD^ (Figure 7F) vs. 87% in ND (Figure 4F). This impaired beta cell identity was prevented in NAC^HFD^ islets (Figure 7D, 7F). In parallel, NAC^HFD^ islets showed significantly reduced 8OHdG and lower intra-islet αSMA labeling compared to ND^HFD^ islets (Figure 7E,G,H).

## 4. Discussion

This is the first in vivo study investigating whether long-term use of the antioxidant NAC benefits glucose metabolism and beta cell function during aging under ND and HFD challenge conditions. We demonstrated that aging mice receiving long-term NAC supplementation displayed significantly lower body weight, improved glucose tolerance, and reduced beta cell oxidative stress, intra-islet PaSC activation, and islet collagen deposition compared to age-matched controls. Interestingly, these mice showed lower fasting plasma insulin and islet insulin content, smaller beta cell mass, and reduced nuclear PDX-1 and insulin exocytosis protein expression compared to age-matched controls (Figure 8). A significant reduction in beta cell oxidative stress and insulin levels in NAC mice was associated with significantly decreased intracellular signaling pathways observed by reduced phosphorylated IRβ^Tyr1146^, IRS-1^Ser612^, Akt^Ser473^, and ERK1/2^Thr202/Tyr204^ (Figure 8), as well as phosphorylated eIF2α^Ser51^ and NFκB. Importantly, long-term NAC supplementation prevented impairments in glucose tolerance, preserved beta cell identity, and lowered intra-islet oxidative stress and PaSC activation in aging mice challenged with HFD stress. This study demonstrates that while long-term NAC administration provides some metabolic benefits in aging, it may have less desirable effects on certain aspects of beta cell function, which may relate to established physiological roles of ROS in this context.

Mice receiving long-term NAC supplementation had consistently lower body weights than their age-matched controls at each studied timepoint. These results are consistent with other studies investigating the anti-obesity effects of daily intraperitoneal or oral NAC administration on male C57BL/6 mice and Sprague–Dawley rats [16,27]. One study suggested that this NAC-induced weight loss was due to a reduction in visceral fat [27], which is similar to the reduced epididymal fat pad weights observed in 60-week NAC mice. It has been shown that reduced body weight allows for the maintenance of glucose regulation and insulin action [28]. Aging mice under long-term NAC showed significantly improved glucose tolerance, even with lower fasting plasma insulin and beta cell insulin content, and no changes to insulin sensitivity, indicating that NAC-induced improvements may be associated with reduced body weight.

It is well established that excess oxidative stress contributes to aging, diabetes, and beta cell dysfunction. However, less is known about the effects of long-term antioxidant-induced reductive stress on aging pancreatic islets. The present study showed that aging mice receiving long-term NAC have significant reductions in beta cell oxidative stress, intra-islet αSMA^+^ cells, and collagen deposition. This observation matches our previous in vivo and other in vitro reports that NAC is capable of reducing PaSC activation and reverting PaSCs to a quiescent state [16,29,30]. We were intrigued to find that the effect of NAC in terms of lowering beta cell 8OHdG (Figure 3B) appeared to correlate with its effect of reducing nuclear PDX-1 (Figure 4F); this may suggest that intracellular ROS balance is necessary to regulate PDX-1 localization and function. Notably, Nrf2 is an important regulator of redox homeostasis under physiological conditions [31]. Furthermore, oxidative stress is involved in the activation of ERK [32,33,34] and PI3K/AKT signaling [35,36,37]. Additionally, ERK and PI3K/AKT signaling have both been shown to induce Nrf2 activation [38,39]. Beta cell Pdx-1 function and/or expression is enhanced by signaling through Nrf2 [40,41,42], ERK [43], and PI3K/Akt [44]. Together, these prior findings suggest that intracellular ROS may promote Nrf2, ERK, and PI3K/AKT signaling to facilitate the maintenance of high beta cell PDX-1 levels. Therefore, in the present study, it seems likely that long-term NAC supplementation in aging mice lowered intracellular ROS levels such that physiological ROS signaling pathways were impaired, which may have contributed to reduced PDX-1 nuclear localization.

The present study demonstrated that NAC mice at 60 weeks had a lower beta cell mass compared to their age-matched controls, with no differences in other aspects of islet morphology, such as alpha cell mass, islet density, and islet distribution. Some evidence suggests that PDX-1 has a direct role in regulating beta cell mass and proliferation in rodents [45]. Therefore, the reduced nuclear PDX-1 expression observed in NAC mice may be at least partially responsible for the lower beta cell mass in NAC-treated mice. A recent study by Ikushima et al. revealed bifunctional roles for ERK signaling in beta cells in the maintenance of glucose homeostasis [46]. They showed that abolishing pancreatic beta cell ERK signaling through deletion of *Mek1* and *Mek2* (β*Mek1/2*DKO) reduced islet area, the number of insulin exocytotic events, and insulin production [46]. This is indicative of an important role in the maintenance of beta cell mass and insulin exocytosis in non-obese mice and is consistent with our present work showing that long-term NAC reduced islet phospho-ERK1/2 expression, beta cell mass, and expression of insulin exocytotic proteins. We postulate that the effect of NAC in terms of lowering ROS also reduced ERK signaling, which may have contributed to these alterations in beta cell biology.

The present study showed that aging mice receiving long-term NAC had reduced expression of insulin exocytotic machinery and reduced insulin signaling. It is proposed that long-term NAC administration lowered oxidative stress levels excessively, perhaps to a state of reductive stress. The present study showed reduced 8OHdG and NFκB levels in NAC mice, the latter of which increases in response to ROS to combat oxidative stress [47,48]. Recent research conducted by Plecitá-Hlavatá et al. suggests that H_2_O_2_ produced by the mitochondria during glucose-stimulated insulin secretion is required for beta cell insulin exocytosis [49]. Therefore, long-term NAC administration may have increased the antioxidative capacity of islets and diminished the levels of ROS required to maintain insulin exocytosis, resulting in less serum insulin and reduced overall insulin secretion during GSIS. Additionally, we observed that mice receiving long-term NAC displayed a significant reduction in phosphorylated eukaryotic translation initiation factor 2 (p-eIF2α^Ser51^), a marker of the integrated stress response. Because oxidative stress can result in ER stress [50], this finding may suggest that the antioxidant actions of NAC potentially made the ER environment less oxidative. Insulin requires an oxidative environment in the ER to achieve proper folding and secretion within beta cells [51,52], and reductive stress can reduce proinsulin levels in MIN6 cells [53]. However, further research needs to be undertaken to explore the connection between NAC, ER stress, and insulin production. This provides supportive evidence of the negative impact antioxidant therapy can have on insulin production and release in beta cells and may provide an explanation for the effect of NAC in terms of reducing islet insulin content and secretion.

Additionally, oxidative stress can affect intracellular insulin signaling. This is dependent on the concentration of oxidants present, where lower concentrations of oxidants promote PI3K/AKT signaling, while high concentrations inhibit this pathway [54]. We found that mice receiving long-term administration of NAC displayed reduced IR/IRS-1/AKT phosphorylation. This is consistent with studies conducted in cultured myotubes and adipocytes exposed to increasing concentrations of NAC, which demonstrated reduced phosphorylation of IR and AKT [55]. The PI3K/AKT signaling pathway regulates SNARE protein levels, where impairing PI3K/AKT signaling results in significantly reduced levels of syntaxin 1A, SNAP25, and VAMP2 in islets [56], and these changes were restored upon expression of constitutively active AKT [57]. In addition to SNARE protein regulation, AKT also regulates PDX-1 expression and associated insulin production, as well as beta cell proliferation and survival [44,58]. The altered expression of SNARE proteins in combination with reduced nuclear PDX-1 expression in mice undergoing long-term NAC supplementation resulted in reduced insulin release and insulin content, which may further affect the IR/IRS/AKT signaling pathway.

Our previous study determined that NAC provided benefits by limiting beta cell overcompensation and loss of beta cell identity typically observed in HFD-induced obese diabetic mice [16]. We also showed that NAC efficacy was dependent on both timing and dosage of NAC intake. Similarly, in the present study, we showed that aging mice receiving long-term NAC while undergoing an HFD-stress challenge had significantly improved glucose tolerance and beta cell function compared to non-NAC-treated HFD mice. Long-term NAC intake was effective at preventing loss of nuclear PDX-1 expression while promoting an apparent increase in membrane localization of GLUT2 and syntaxin 1A. This suggests that NAC may reduce hyperinsulinemia in aging mice undergoing an HFD-stress challenge by maintaining membrane localization of GLUT2 and SNARE proteins. Additionally, long-term NAC prevents HFD-induced beta cell oxidative stress and intra-islet PaSCs activation, which is consistent with our previous report [16]. Therefore, this HFD-stress challenge study further verified the functional role of NAC on pathological conditions that NAC provides a protective effect to aging mice to overcome oxidative stress-induced beta cell dysfunction.

A limitation of this study is that mice received NAC by administration ad libitum in the drinking water. Although this resulted in consistent daily NAC consumption, this continuous administration of NAC is distinct from the route of administration used in clinical studies, in which NAC is consumed orally in the form of capsules. Therefore, in future studies, it will be important to evaluate whether the effects of NAC observed in the present study are reproducible when mice are dosed using a more clinically relevant route of administration. Additionally, this study was conducted on aging mice (~6–14 months), likely experiencing no or relatively mild metabolic impairments and lower age-related oxidative stress compared to their aged (>18 months) counterparts [59,60]. As previously mentioned, the aim of this study was to investigate the long-term dietary supplementation of NAC in aging mice and its impact on beta cell function and insulin signaling. NAC may have had more of a pronounced effect on reducing oxidative stress, resulting in “reductive stress”, which resulted in the metabolic and beta cell-specific changes observed with long-term NAC administration in this study. Aging individuals (over 65 years old) are susceptible to developing type 2 diabetes, and these people present with insulin resistance that eventually leads to defects in insulin secretion. Indeed, insulin resistance, increased basal insulin levels, and beta cell dysfunction are all observed as a part of aging in C57BL/6 mice [60,61]. Future studies should also investigate the use of NAC in aged mice (>18 months) to determine the effects of NAC on mice with more severe age-related metabolic impairments to determine if there is benefit in its use for the treatment of age-related pathologies, such as diabetes.

## 5. Conclusions

In summary, the present study demonstrates that long-term NAC administration in normal-diet-fed aging mice may have excessively lowered beta cell intracellular ROS levels, which may have contributed to several changes we observed, including reductions in PI3K/AKT and ERK signaling, PDX-1 nuclear localization, expression of insulin exocytotic proteins, and insulin production and secretion (Figure 8). The effects of reductive stress on insulin signaling have been studied in other tissues such as adipose and muscle. This is the first in vivo study investigating how NAC affects insulin signaling within islets. Furthermore, this work highlights the need to further investigate ROS balance in beta cell biology to determine if antioxidant therapies can successfully reduce pathological oxidative stress without perturbing physiological pathways.

## Figures and Tables

**Figure 1 antioxidants-14-00417-f001:**
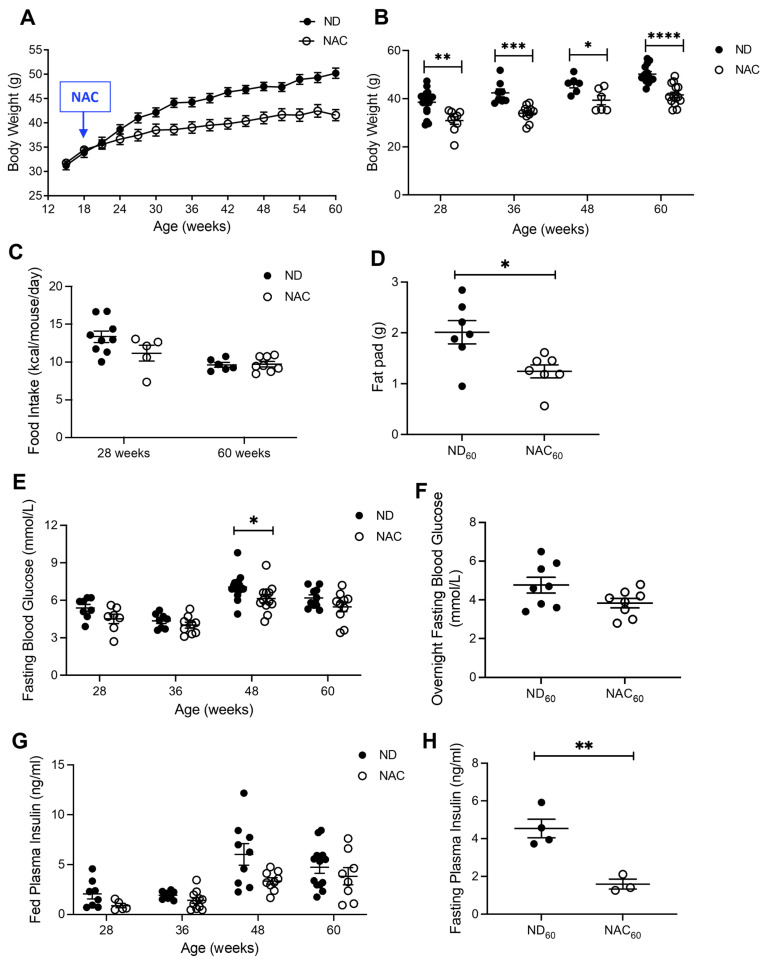
Long-term NAC supplementation during aging displays lean body mass, low fat pad mass, and reduced fasting plasma insulin levels. (**A**) Body weight trend recording for the duration of 60 weeks (*n* = 13 ND; *n* = 14 NAC mice/group). (**B**) Measurement of final body weight at 28 weeks (*n* = 17 ND; *n* = 9 NAC mice/group), 36 weeks (*n* = 8 ND; *n* = 11 NAC mice/group), 48 weeks (*n* = 6 ND; *n* = 6 NAC mice/group), and 60 weeks (*n* = 13 ND; *n* = 14 NAC mice/group). (**C**) Food intake measurements at 28 weeks (*n* = 9 ND; *n* = 5 NAC mice/group) and 60 weeks (*n* = 6 ND; *n* = 8 NAC mice/group). (**D**) Fat pad weight at 60 weeks (*n* = 7 mice/group). (**E**) Fasting (4 h) blood glucose at 28 weeks (*n* = 8 ND; *n* = 7 NAC mice/group), 36 weeks (*n* = 8 ND; *n* = 9 NAC mice/group), 48 weeks (*n* = 10 ND; *n* = 6 NAC mice/group), and 60 weeks of age (*n* = 8 ND; *n* = 7 NAC mice/group). (**F**) Overnight fasting blood glucose at 60 weeks (*n* = 8 mice/group). (**G**) Fed plasma insulin at 28 weeks (*n* = 8 ND; *n* = 5 NAC mice/group), 36 weeks (*n* = 8 ND; *n* = 11 NAC mice/group), 48 (*n* = 9 ND; *n* = 3 NAC mice/group), and 60 weeks (*n* = 6 mice/group). (**H**) Fasting plasma insulin at 60 weeks (*n* = 4 ND; *n* = 3 NAC mice/group). Control diet (ND): closed circle; NAC: open circle. Data are expressed as means ± SEM. * *p* < 0.05, ** *p* < 0.01, *** *p* < 0.001, **** *p* < 0.0001; analyzed using unpaired Student’s *t*-tests.

**Figure 2 antioxidants-14-00417-f002:**
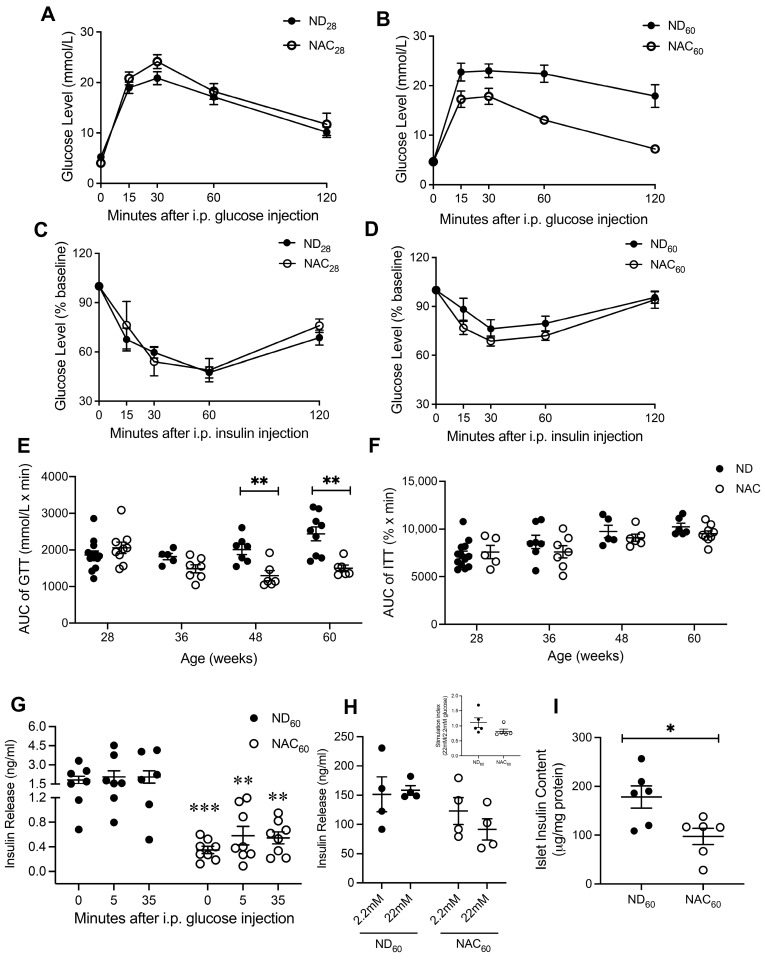
Long-term NAC supplementation improves glucose tolerance with low islet insulin content during aging. IPGTT at 28 weeks ((**A**), *n* = 11 ND; *n* = 9 NAC mice/group) and at 60 weeks ((**B**), *n* = 9 ND; *n* = 6 NAC mice/group). IPITT at 28 weeks ((**C**), *n* = 13 ND; *n* = 5 NAC mice/group) and at 60 weeks ((**D**), *n* = 6 ND; *n* = 9 NAC mice/group). (**E**) AUC of IPGTT at 28 weeks (*n* = 11 ND; *n* = 9 NAC mice/group), 36 weeks (*n* = 5 ND; *n* = 7 NAC mice/group), 48 weeks (*n* = 7 ND; *n* = 6 NAC mice/group), and 60 weeks (*n* = 9 ND; *n* = 6 NAC mice/group). (**F**) AUC of IPITT at 28 weeks (*n* = 13 ND; *n* = 5 NAC), 36 weeks (*n* = 7 ND; *n* = 7 mice/group), 48 weeks (*n* = 5 ND; *n* = 6 NAC mice/group), and 60 weeks (*n* = 6 ND; *n* = 9 NAC mice/group). (**G**) In vivo GSIS at 60 weeks (*n* = 7 ND; *n* = 8 mice/group); statistics were performed using unpaired Student’s *t*-tests within each timepoint. (**H**) Ex vivo GSIS in isolated islets at 60 weeks (*n* = 4 mice/group). (**I**) Isolated islet insulin content at 60 weeks (*n* = 6 mice/group). Control diet (ND): closed circle; NAC: open circle. Data are expressed as means ± SEM. * *p* < 0.05, ** *p* < 0.01, *** *p* < 0.001; analyzed using unpaired Student’s *t*-tests.

**Figure 3 antioxidants-14-00417-f003:**
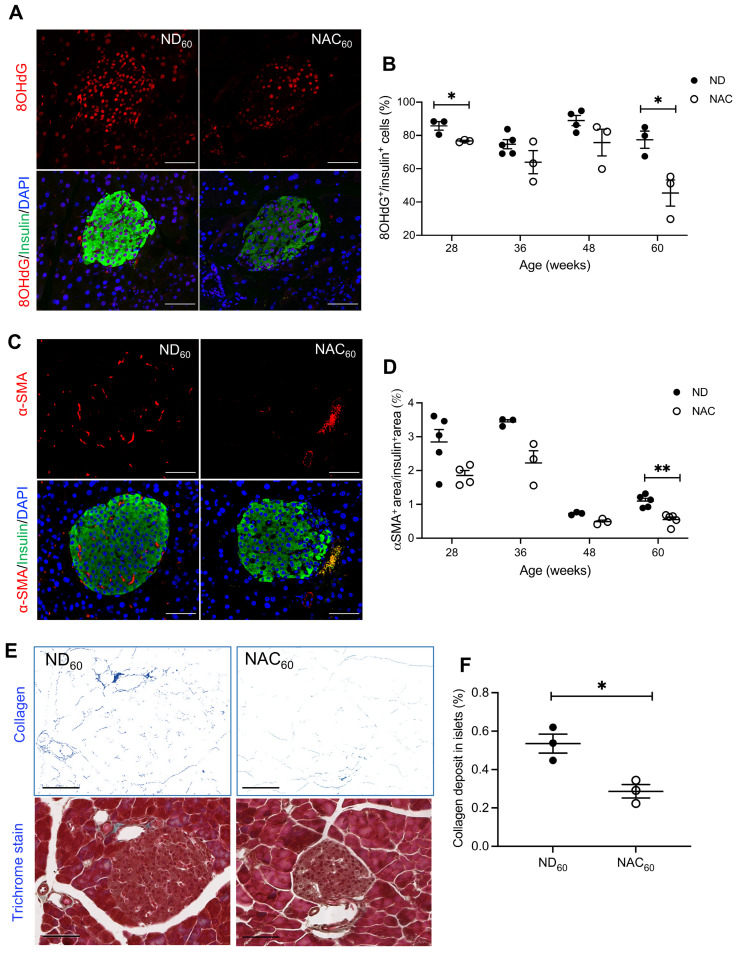
Long-term NAC supplementation reduces beta cell oxidative stress, intra-islet PaSCs activation and collagen deposition in aging mice. Representative images of double immunofluorescence of (**A**) 8OHdG and (**C**) α-SMA (red) co-stained with insulin (green); nuclei are stained with DAPI (blue). Scale bars: 50 µm. Quantification of (**B**) 8OHdG in insulin^+^ cells at 28 weeks (*n* = 3 pancreata/group), 36 weeks (*n* = 5 ND; *n* = 3 NAC pancreata/group), 48 weeks (*n* = 4 ND; *n* = 3 NAC pancreata/group), and 60 weeks (*n* = 3 pancreata/group). Quantification of (**D**) α-SMA within insulin^+^ area of mouse pancreata at 28 weeks (*n* = 5 ND; *n* = 4 NAC pancreata/group), 36 weeks (*n* = 3 pancreata/group), 48 weeks (*n* = 3 NAC pancreata/group), and 60 weeks (*n* = 5 ND; *n* = 4 NAC pancreata/group). (**E**) Representative trichrome staining images, scale bar: 50 μm; and (**F**) quantification of intra-islet collagen deposition (*n* = 3 pancreata/group). Control diet (ND): closed circle; NAC: open circle. Data are expressed as means ± SEM. * *p* < 0.05, ** *p* < 0.01; analyzed using unpaired Student’s *t*-tests.

**Figure 4 antioxidants-14-00417-f004:**
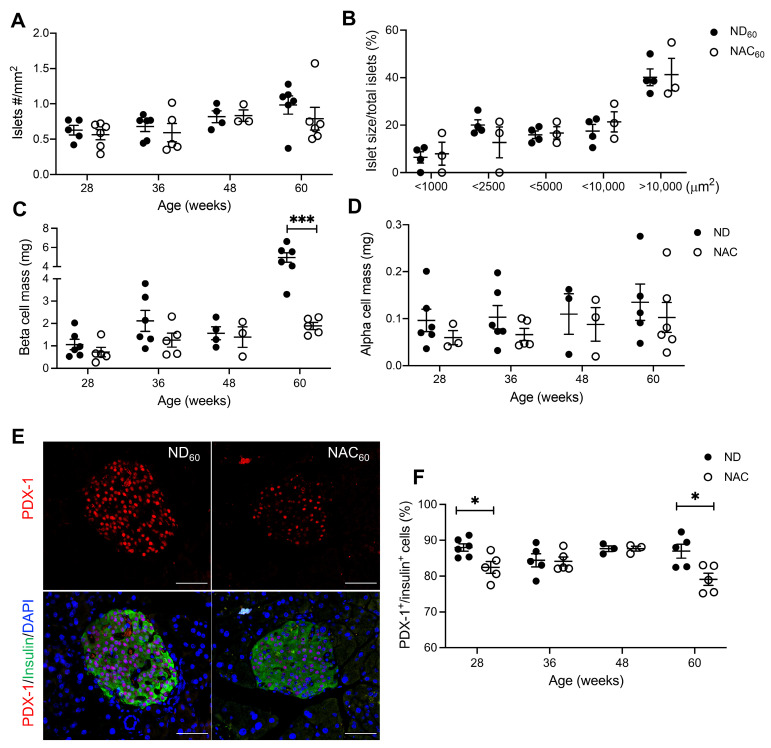
Long-term NAC supplementation preserves beta cell mass but reduces nuclear PDX-1 expression in aging mice. (**A**) Islet density at 28 weeks (*n* = 5 ND; *n* = 6 NAC mice/group), 36 weeks (*n* = 6 ND, *n* = 5 NAC mice/group), 48 weeks (*n* = 4 mice/group), and 60 weeks (*n* = 6 mice/group). (**B**) Islet size distribution at 60 weeks of age (*n* = 4 ND; *n* = 3 NAC mice/group). Islet morphometric analysis of (**C**) beta cell mass at 28 weeks (*n* = 6 ND; *n* = 5 NAC mice/group), 36 weeks (*n* = 6 ND; *n* = 5 NAC mice/group), 48 weeks (*n* = 4 ND; *n* = 3 NAC mice/group), and 60 weeks (*n* = 5 ND; *n* = 6 NAC mice/group). (**D**) Alpha cell mass at 28 weeks (*n* = 6 ND; *n* = 3 NAC mice/group), 36 weeks (*n* = 6 ND; *n* = 5 NAC mice/group), 48 weeks (*n* = 3 mice/group), and 60 weeks (*n* = 5 ND; *n* = 6 NAC mice/group). (**E**) Representative double immunofluorescence images of PDX-1 (red) co-stained with insulin (green); nuclei are stained with DAPI (blue). Scale bars: 50 µm. (**F**) Quantification of nuclear PDX-1 in insulin^+^ cells at 28 weeks (*n* = 6 ND; *n* = 5 NAC pancreata/group), 36 weeks (*n* = 5 pancreata/group), 48 weeks (*n* = 3 pancreata/group), and 60 weeks (*n* = 5 pancreata/group). Control diet (ND): closed circle; NAC: open circle. Data are expressed as means ± SEM. * *p* < 0.05, *** *p* < 0.001; analyzed using unpaired Student’s *t*-tests.

**Figure 5 antioxidants-14-00417-f005:**
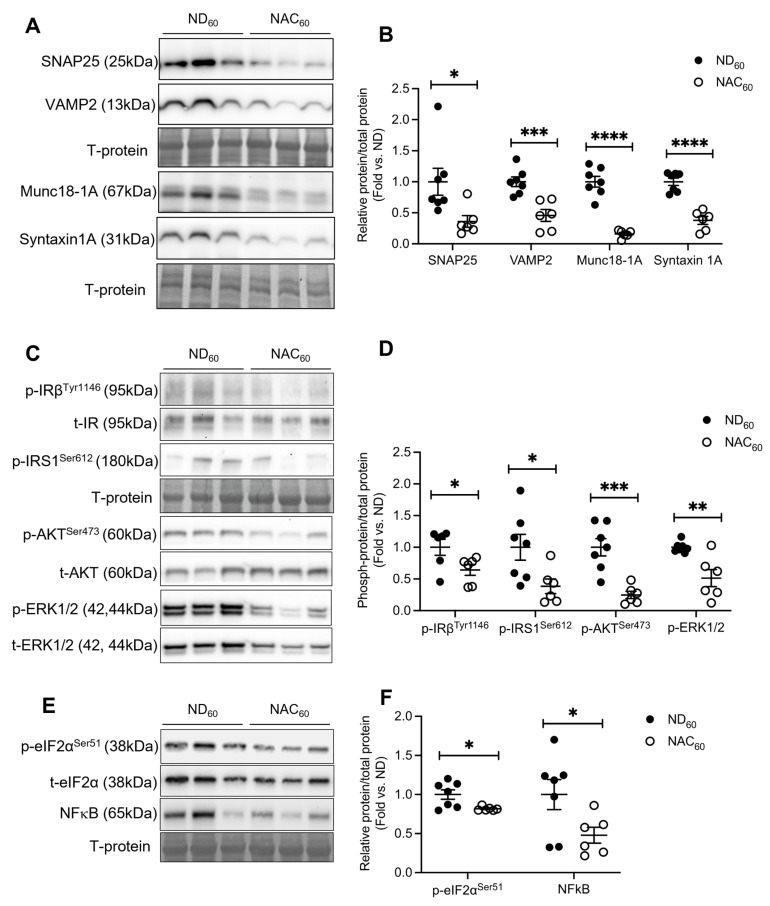
Long-term NAC supplementation reduced ROS and altered insulin exocytotic protein levels and intracellular signaling pathways. (**A**) Representative Western blot images and (**B**) densitometry quantification of SNARE proteins SNAP25, VAMP2, Munc18-1A, and syntaxin 1A in isolated mouse islets at 60 weeks (*n* = 7 ND; *n* = 6 NAC mouse-isolated islets/group). (**C**) Representative Western blot images and (**D**) densitometry quantification of phosphorylated proteins IRβ^Tyr1146^, IRS-1^Ser612^, Akt^Ser473^, and ERK1/2, and their total proteins (*n* = 7 ND; *n* = 6 NAC mouse-isolated islets/group). (**E**) Representative Western blot and (**F**) densitometry quantification of phosphorylated and total eIF2α and NFκB (*n* = 7 ND; *n* = 6 NAC mouse-isolated islets/group). Control diet (ND): closed circle; NAC: open circle. Data are expressed as means ± SEM. * *p* < 0.05, ** *p* < 0.01, *** *p* < 0.001, **** *p* < 0.0001; analyzed using unpaired Student’s *t*-tests.

**Figure 6 antioxidants-14-00417-f006:**
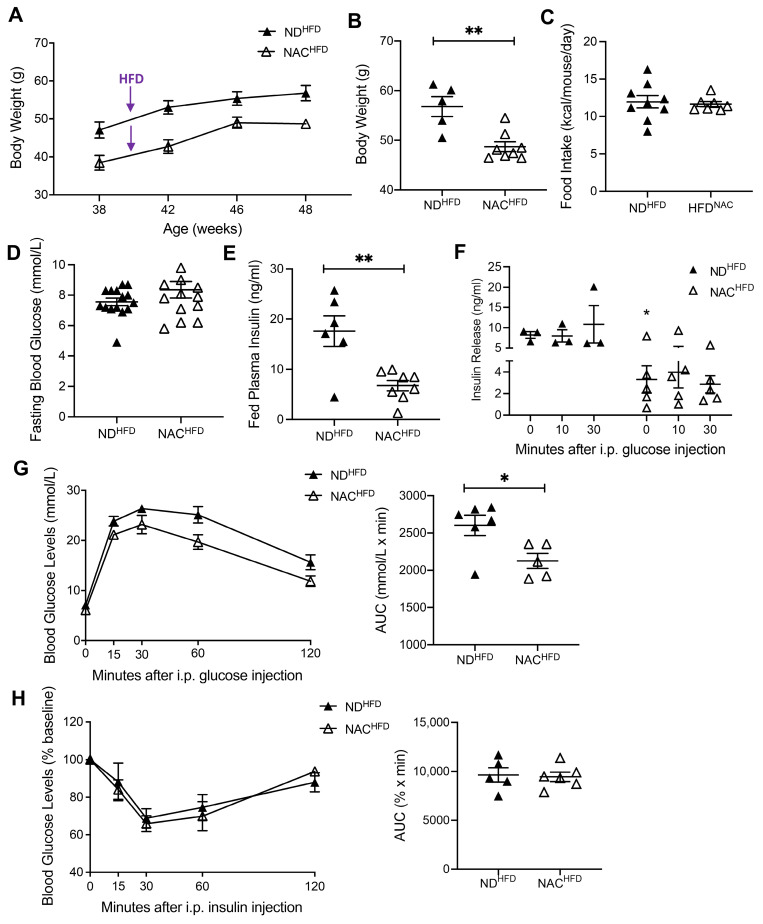
Long-term NAC supplementation preserves weight and glucose metabolism in aging mice undergoing HFD challenge. (**A**) Body weight trends during HFD challenge (*n* = 5 ND^HFD^; *n* = 8 NAC^HFD^ mice/group). Body weight (**B**) (*n* = 5 ND^HFD^; *n* = 8 NAC^HFD^ mice/group), food intake measurements (**C**) (*n* = 7 ND^HFD^; *n* = 9 NAC^HFD^ mice/group), fasting blood glucose (**D**) (*n* = 15 ND^HFD^; *n* = 12 NAC^HFD^ mice/group), and fed plasma insulin (**E**) (*n* = 6 ND^HFD^; *n* = 8 NAC^HFD^ mice/group) after 8 weeks of HFD challenge. (**F**) In vivo GSIS after 8 weeks HFD challenge at 48 weeks of age (*n* = 3 ND^HFD^; *n* = 5 NAC^HFD^ mice/group); statistics were performed using unpaired Student’s *t*-test within each timepoint. (**G**) IPGTT and AUC (*n* = 6 ND^HFD^; *n* = 5 NAC^HFD^ mice/group) and (**H**) IPITT and AUC (*n* = 5 ND^HFD^; *n* = 6 NAC^HFD^ mice/group) after 8 weeks of HFD challenge. HFD challenge in control diet (ND^HFD^): closed triangle; HFD challenge in NAC treatment group (NAC^HFD^): open triangle. Data are expressed as means ± SEM. * *p* < 0.05, ** *p* < 0.01; analyzed using unpaired Student’s *t*-tests.

**Figure 7 antioxidants-14-00417-f007:**
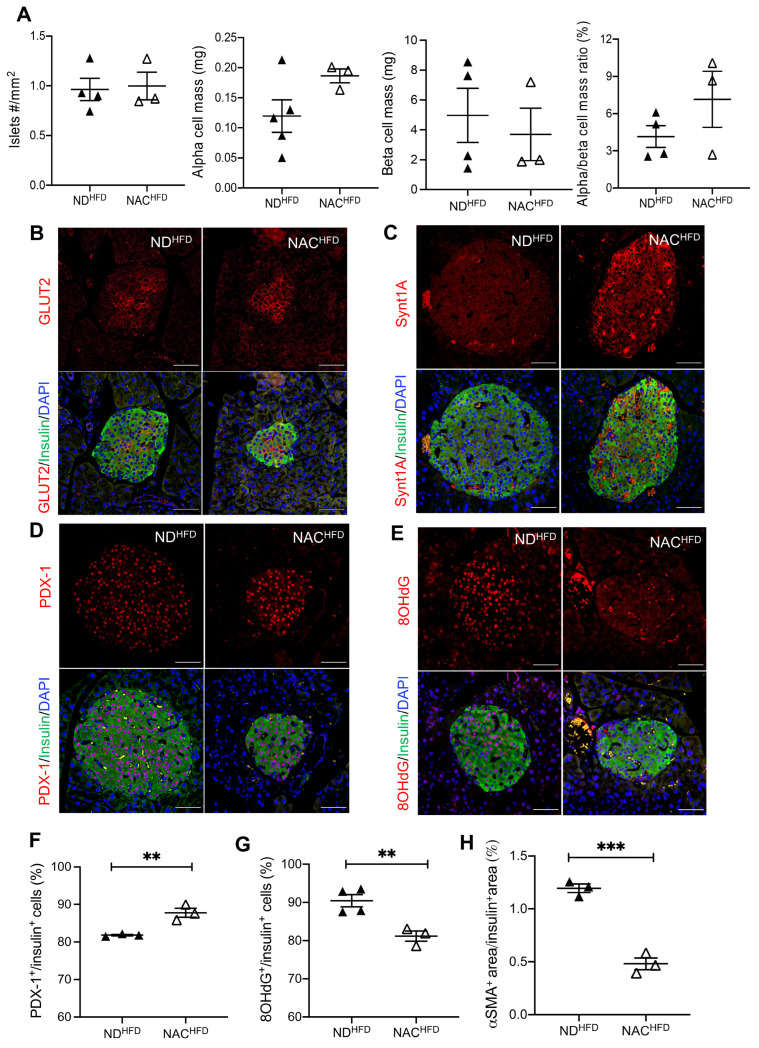
Long-term NAC supplementation preserves nuclear PDX-1 expression and reduces beta cell oxidative stress in aging mice undergoing HFD challenge. (**A**) Quantification of islet density, alpha cell mass, beta cell mass, and alpha/beta cell mass ratio in immunofluorescent images from ND and NAC mice 8 weeks after HFD challenge. Representative double immunofluorescence images for (**B**) GLUT2, (**C**) syntaxin 1A, (**D**) PDX-1, and (**E**) 8OHdG labeled in red co-stained with insulin (green); nuclei are stained with DAPI (blue). Scale bars: 50 µm. Quantification of (**F**) PDX-1, (*n* = 3 pancreata/group), (**G**) 8OHdG (*n* = 4 ND^HFD^; *n* = 3 NAC^HFD^ pancreata/group), and (**H**) αSMA (*n* = 3 pancreata/group). HFD challenge in control diet (ND^HFD^): closed triangle; HFD challenge in NAC treatment group (NAC^HFD^): open triangle. Data are expressed as means ± SEM. ** *p* < 0.01, *** *p* < 0.001; analyzed using unpaired Student’s *t*-tests.

**Figure 8 antioxidants-14-00417-f008:**
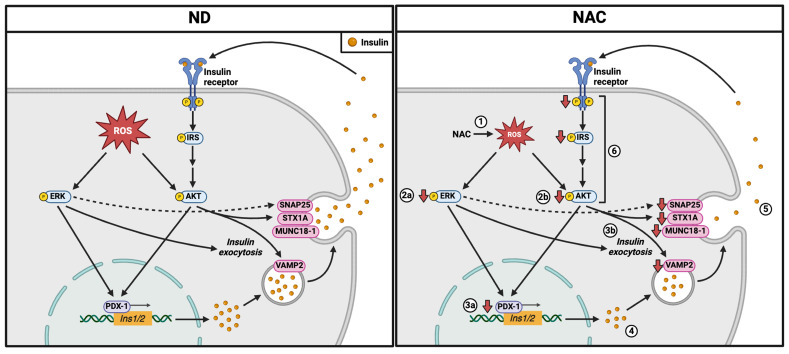
Schematic for proposed model of the effects of long-term antioxidant NAC therapy on beta cell function in aging mice. In normal diet (ND) conditions (**left**), physiological levels of reactive oxygen species (ROS) in beta cells promote phosphorylation of AKT and ERK signaling, which promote PDX-1 activity. PDX-1 promotes transcription of the insulin gene, resulting in insulin synthesis and secretion. ROS and insulin-linked AKT pathway enhance the expression of proteins involved in insulin exocytosis, while ROS-linked ERK pathway promotes insulin exocytosis, perhaps through phosphorylation of SNAP25. In long-term NAC treatment (**right**), **(1)** NAC reduces intracellular ROS levels, which reduces phosphorylation of **(2a)** ERK1/2 and **(2b)** AKT, which reduces **(3a)** nuclear translocation of PDX-1 and **(3b)** expression of insulin exocytotic proteins. Reduced nuclear PDX-1 results in **(4)** less insulin production. Lower insulin content and reduced expression of insulin exocytosis proteins **(5)** decrease insulin secretion. This results in **(6)** less autocrine insulin signaling and **(2b)** further reduces AKT phosphorylation. Created in BioRender.com.

## Data Availability

The datasets presented in this article are not readily available because the data are part of an ongoing study to access the datasets should be directed to the corresponding author, rwang@uwo.ca.

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
