# Peer review of "Long-Term Administration of Antioxidant N-Acetyl-L-Cysteine Impacts Beta Cell Oxidative Stress, Insulin Secretion, and Intracellular Signaling Pathways in Aging Mice"

_antioxidants, 2025, doi:10.3390/antiox14040417_

Round 1

Reviewer 1 Report

This is an exciting study where Schuurman and colleagues explore the long-term effects of N-acetyl-L-cysteine (NAC) on glucose metabolism and beta-cell function in aging mice under normal and high-fat diet (HFD) conditions. NAC supplementation reduced body weight, fat mass, and beta-cell oxidative stress while improving glucose tolerance and lowering intra-islet pancreatic stellate cell  activation and islet fibrosis. However, it decreased fasting insulin levels, beta-cell mass, insulin exocytotic protein expression, and nuclear localization of PDX-1, a key transcription factor. NAC also impaired insulin signaling pathways by reducing AKT and ERK1/2 phosphorylation, possibly due to excessive reduction of reactive oxygen species (ROS). Under HFD stress, NAC preserved beta-cell identity and function by reducing oxidative stress and maintaining critical protein localization. These findings suggest that while NAC mitigates oxidative stress and metabolic dysfunction, it may interfere with essential physiological processes, highlighting the need for a balanced approach to antioxidant therapy in aging and metabolic disease. Although these findings are very interesting, crucial points need clarification, and some experimental comparisons are needed before the current study is ready for publication

  1. The authors utilize NAC at 50 mM, which is a relatively high dose for in vivo studies. What are the potential impacts of this dose on the mice? How does this concentration compare to doses used in the literature? A discussion on the rationale for selecting this dose and any potential side effects would improve the manuscript. To my knowledge many studies use 1g/L NAC which is equivalent to 6.14 mM NAC, and to my experience this is the NAC concentration point at which the drinking water gets sour (low pH) and would need to be sweetened (which can vary depending on the alkalinity of the local drinking water). Please clarify, and provide rationale and explanations. What is the pH of the 50mM NAC water used? Do the authors modify this in any additional way (adding sugar?). If the authors have not modified the drinking water in anyway, please try tasting 50mM NAC water, perfectly safe but very sour.

2       Age of the mice and its translational relevance: The oldest time point studied is 60 weeks (≈14 months), which corresponds to <50 human years (Ref: Flurkey et. al, 2004; Flurkey et. al 2007) or the Mouse in Biomedical Research.

a)How does this relate to the onset of type 2 diabetes (T2D), insulin resistance, and glucose intolerance in aging humans? Clarifying this would help assess the relevance of the model. The authors should explain how the 60-weeks of age has been chosen.

b) Although T2D onset occurs around 45+ years in humans, its incidence increases significantly after 60 years (https://www.cdc.gov/diabetes/php/data-research/index.html). Therefore, a limitation of the study is that NAC’s effects may be pronounced because metabolic impairments at this age (max 60 weeks) are still relatively mild. The authors need to discuss this point.

3       Comparison of HFD +/- NAC  with ND 48 weeks (no HFD) is lacking. This comaprison is necesssry to fully apprehend the extent of i)HFD with aging; ii) NAC rescue effect on HFD with aging.

4    In line with previous comment, the rationale for using HFD and studying NAC effect on HFD should be better explained.

  1. The authors are using pEIF2a as a marker of ER stress (line 262-263). However, EIF2a is a broader marker of the integrated stress response (ISR), and not exclusiverly of ER stress. Therefore, the conclusions to this observations (reduced p-EIF2a) should be corrected accordingly (lines 264-267). If the authors want to comment on ER stress they need to evaluate levels of the ER stress branches which include IRE1a -> XBP1 splicing and ATF6 levels respectively.

Thus, If the authors want to check and make conclusions on ER stress status, which is relevant in the context of T2D, other more specific markers must be used.

  1. Pathways investigated in Figure 5 and shown as decreased signaling in NAC-treated mice (ND60) should also be investigated in HFD; does the HFD increase these pathways (Fig5A SNARE exocytotic pathway, and 5C Insulinreceptor signaling and AKT/ERK) which were rescued by NAC?

  1. Inconsistency in sample size across figures: the number of mice for a given experimental group varies across different panels of the same figure. Example: Figure 1B (body weight, ND60, n=6), Figure 1C & 1D (ND60, n=7), and Figure 1F (ND60, n=8). Why the group for body weight measurement (the less invasive measurement) is smaller than other more invasive measurements? If some analyses were conducted on a subset of animals, this should be clearly stated in the Methods, currently key information missing.

  1. It is currently unclear if the authors have conducted one experiment or multiple experiments with long term NAC treatment. The n = needs be defined in every figure and cannot be a range!
  2. As the authors have conducted a 60week long study on C57/B6 animals, and stated that “individuals with health issues were excluded” this need to be better reported especially as it is a longitudinal study and potential adverse effects of NAC treatment or HFD treatment might be selected away from the final analysis. Indeed it is well known that in the pure C57/B6 background animals tend to get spontaneous tumors and lymphomas around 1-year, and it wouldn´t be surprising if HFD and or NAC might impact the outcome of malignancies in such cohorts in light of a plethora of research on NAC/antioxidants and HFD and their impact on tumor progression in the last decade...

Minor comments:

  1. Legend Figure 1; mentioning biweekly weight measurement. Based on graph shown Figure 1A showimg measurement every other week, the authors meant “every other week” which is not the same as two-times every week (4-fold difference)

Author Response

RESPONSE TO REVIEWER #3 comments

Comment #1

The authors utilize NAC at 50 mM, which is a relatively high dose for in vivo studies. What are the potential impacts of this dose on the mice? How does this concentration compare to doses used in the literature? A discussion on the rationale for selecting this dose and any potential side effects would improve the manuscript. To my knowledge many studies use 1g/L NAC which is equivalent to 6.14 mM NAC, and to my experience this is the NAC concentration point at which the drinking water gets sour (low pH) and would need to be sweetened (which can vary depending on the alkalinity of the local drinking water). Please clarify, and provide rationale and explanations. What is the pH of the 50mM NAC water used? Do the authors modify this in any additional way (adding sugar?). If the authors have not modified the drinking water in anyway, please try tasting 50mM NAC water, perfectly safe but very sour.

This methodology was approved in our animal use protocol within guidelines of Canadian Council of Animal Care. The concentrations used in our work compare well with a published study by (Ref: Falach-Malik et al., 2016) where they found ~600-1200mg/kg/day had the best metabolic outcomes. Animals in our study consumed NAC within this range. Our 2022 study also used 10mM drinking water and we found that this concentration was not sufficient to improve metabolic outcomes. We did test the pH in 10mM and 50 mM NAC solutions and found that they are both pH 2.3. We did not modify NAC containing drinking water with sugar or other sweetener for this study. The mice drank the NAC-supplemented water, with no adverse health reports or any dehydration noted by our facility animal care staff or researchers due to NAC consumption.

Comment #2

Age of the mice and its translational relevance: The oldest time point studied is 60 weeks (≈14 months), which corresponds to <50 human years (Ref: Flurkey et. al, 2004; Flurkey et. al 2007) or the Mouse in Biomedical Research.

  1. How does this relate to the onset of type 2 diabetes (T2D), insulin resistance, and glucose intolerance in aging humans? Clarifying this would help assess the relevance of the model. The authors should explain how the 60-weeks of age has been chosen.

This study was conducted to determine the effects of long-term NAC during healthy aging mice. Based on our previous published work (Oakie et al., 2019 Diabetologia) and in initiating our current aging study, we found that mice after 60 weeks age begin to develop adverse age-related health outcomes such as reduced survival, spontaneous tumour development etc. In order to maintain sufficient number of mice for the current study, we limited the study duration to 60 weeks.

  1. Although T2D onset occurs around 45+ years in humans, its incidence increases significantly after 60 years (https://www.cdc.gov/diabetes/php/data-research/index.html). Therefore, a limitation of the study is that NAC’s effects may be pronounced because metabolic impairments at this age (max 60 weeks) are still relatively mild. The authors need to discuss this point.

  • As mentioned in (a) above, this study was designed to investigate the effects of long-term NAC treatment in healthy aging mice, not in aged diabetic mice. We also examined whether long-term NAC treatment can prevent fat-induced stress leading to the impairment of glucose tolerance and beta-cell function, using a HFD challenge during aging.
  • Regarding aging related to impaired glucose metabolism, from Figure 1E, you can see modest impairments of glucose tolerance over time, indicated by a trend towards increased AUC of GTT. However, mice receiving NAC appear to have reduced AUC of GTT overtime indicated improved glucose tolerance. We respectfully disagree that the effects of NAC may be more pronounced at this age as numerous studies show significant impacts on GTT and ITT in diabetic and/or severely obese mice which are significantly improved with NAC treatment, as reported in our previous work and by other researchers.

Comment #3

Comparison of HFD +/- NAC with ND 48 weeks (no HFD) is lacking. This comparison is necessary to fully apprehend the extent of

i)HFD with aging; ii) NAC rescue effect on HFD with aging.

Although direct comparisons were not made between ND, NDHFD and NACHFD at 48 weeks, all of the ND 48 weeks data is already included in this manuscript in the aging time course branch of the study. This portion of the study was conducted to determine if NAC could rescue negative metabolic outcomes associated with a HFD-challenge during aging. Comparisons between NDHFD and NACHFD achieves this goal.

Comment #4

In line with previous comment, the rationale for using HFD and studying NAC effect on HFD should be better explained.

We determined with our 2022 published study that NAC is beneficial as a prevention and intervention for improving negative metabolic outcomes associated with long-term HFD feeding (>20 weeks), which was conducted in mice from 6 to 28/36 weeks of age. The HFD challenge in the current manuscript was conducted as an extension to our previous study, to determine if NAC was effective for prevention of negative metabolic outcomes associated with a HFD challenge during aging.

Comment #5

The authors are using pEIF2a as a marker of ER stress (line 262-263). However, EIF2a is a broader marker of the integrated stress response (ISR), and not exclusively of ER stress. Therefore, the conclusions to this observations (reduced p-EIF2a) should be corrected accordingly (lines 264-267). If the authors want to comment on ER stress they need to evaluate levels of the ER stress branches which include IRE1a -> XBP1 splicing and ATF6 levels respectively.

Thus, If the authors want to check and make conclusions on ER stress status, which is relevant in the context of T2D, other more specific markers must be used.

Thank you for this comment, we have updated to the manuscript to reflect pEIF2a as a marker of the integrated stress response. This update is reflected in line 265, 403-404.

Comment #6

Pathways investigated in Figure 5 and shown as decreased signaling in NAC-treated mice (ND60) should also be investigated in HFD; does the HFD increase these pathways (Fig5A SNARE exocytotic pathway, and 5C Insulin receptor signaling and AKT/ERK) which were rescued by NAC?

Islet isolations were only conducted on the 60 week old mice in this study. For the HFD-challenge at 48 weeks, we performed in vivo glucose metabolism tests and collected pancreatic tissue for the immunofluorescence staining of mouse islets. We agree it would be interesting to investigate how long-term NAC impacts these pathways during HFD stress, and it is a goal of our current in vitro research. It was also not feasible to preserve the 3 Rs of animal research (as required by our Canadian Council on Animal Care), particularly “reduce,” we would have had to approximately triple the number of mice used in the 48 weeks age study in order to achieve the necessary n-value.

Comment #7

Inconsistency in sample size across figures: the number of mice for a given experimental group varies across different panels of the same figure. Example: Figure 1B (body weight, ND60, n=6), Figure 1C & 1D (ND60, n=7), and Figure 1F (ND60, n=8). Why the group for body weight measurement (the less invasive measurement) is smaller than other more invasive measurements? If some analyses were conducted on a subset of animals, this should be clearly stated in the Methods, currently key information missing.

Thank you for drawing this to our attention. The body weight trends and final body weight graphs have been updated in Figure 1A and 1B.

Comment #8

It is currently unclear if the authors have conducted one experiment or multiple experiments with long term NAC treatment. The n = needs be defined in every figure and cannot be a range!

Due to the varied n-values across our timepoints, we included individual data points on our graphs to be transparent with readers and so n-values can be observed in each data set.

Comment #9

As the authors have conducted a 60 week long study on C57/B6 animals, and stated that “individuals with health issues were excluded” this need to be better reported especially as it is a longitudinal study and potential adverse effects of NAC treatment or HFD treatment might be selected away from the final analysis. Indeed it is well known that in the pure C57/B6 background animals tend to get spontaneous tumors and lymphomas around 1-year, and it wouldn´t be surprising if HFD and or NAC might impact the outcome of malignancies in such cohorts in light of a plethora of research on NAC/antioxidants and HFD and their impact on tumor progression in the last decade...

“Health issues” in this context referred to issues outside of study conditions, such as animal fighting and dermatitis lesions due to over barbering which led to weight changes, and not due to study conditions. We did not investigate the effects of NAC on tumors in this study and cannot comment on that in the context of this study. We did not observe negative health outcomes due to NAC or HFD administration. We would like to emphasize again, any mice that developed health concerns did not develop these concerns as a result of study conditions, and were excluded from the study.

Minor Comment #1

Legend Figure 1; mentioning biweekly weight measurement. Based on graph shown Figure 1A showing measurement every other week, the authors meant “every other week” which is not the same as two-times every week (4-fold difference)

Biweekly has dual meaning, either every other week or twice weekly. We agree it is an ambiguous terminology. We have updated the figure legend to remove any confusion.

Reviewer 2 Report

This article does not present new results or conclusions considering the previously published by the same research group. In addition, it is difficult to correlate a lower sensitivity for the secretion of insulin in the treated animals with better glucose homeostasis in the animals.

Comments are indicated above regarding methodology and results normalization.

Author Response

RESPONSE TO REVIEWER #2 comments.

For this reviewer to allege that the study design (minus a few changes), content and conclusions are the same as our previously published study: “N-acetyl-L-cysteine treatment reduces beta-cell oxidative stress and pancreatic stellate cell activity in a high-fat diet-induced diabetic mouse model,” raises concern that they did not understand our current submitted study and did not thoroughly read either study.

We would like first to emphasize the significance and novelty of our current study here:

What is already known about this subject?

  • Aging is an associated risk for the development of T2D, and the aging islet has increased oxidative stress, PaSCs activation and fibrosis, leading to beta-cell dysfunction.
  • While oxidative stress can be detrimental to beta-cells, their low antioxidant capacity appears to be necessary to allow low concentrations of oxidative stress for beta-cell function and insulin release.
  • The antioxidant N-acetyl-L-cysteine (NAC) can prevent the activation of PaSCs and maintain glucose tolerance and insulin sensitivity in mice fed a HFD, demonstrating a potential beneficial role for antioxidants in the diet-induced islet stress.

What is the key question?

  • Does the long-term administration of NAC improve the beta-cell dysfunction associated with aging and diet-induced changes during aging?

What are the new findings?

  • Long-term NAC administration can prevent the development of age-related glucose intolerance, reduce islet oxidative stress and intra-islet PaSCs activation.  
  • While long-term NAC treatment, in aging mice fed chow diet, resulted in reduced beta-cell nuclear PDX-1 expression, insulin signalling, and insulin exocytotic protein expression, which contribute to lower insulin release from beta-cells.
  • Long-term NAC treatment rescued glucose intolerance, restored nuclear PDX-1 expression, and promoted membrane GLUT2 and insulin exocytotic protein expression in mice fed a high-fat diet during aging.

How might this impact on clinical practice in the foreseeable future?

  • This study suggests that the administration of antioxidant therapies for managing glucose intolerance and beta-cell dysfunction must be tightly regulated to prevent any complications from long-term administration.

Importantly, this study is the first to examine how long-term antioxidant NAC impacts the islet microenvironment and insulin signaling of aging islets. Our work demonstrates the importance of studying the role of ROS signaling in islets under physiological conditions. Understanding the role antioxidants play in islet signaling during normal physiology is important to determine when it should be applied without interfering with essential physiological processes.

To respond the current study same as our 2022 published study, here we outlined our findings from Schuurman et al., 2022 and findings from current study

The key findings from Schuurman et al., 2022 are as follows:

Prevention treatment and intervention treatment with NAC in HFD-induced obesity and diabetic mice (up to 30 weeks of HFD feeding, see attached experimental timeline 2022 Frontiers in Endocrinology) was beneficial for maintenance of healthy beta-cells and quiescent intra-islet pancreatic stellate cells, however, the efficiency of NAC treatment is dependent on both timing and dosage. This study was on a diet-induced obesity and diabetes model, and not an aging mouse with normal diet and undergoing a HFD challenge model.

In contrast, the key findings from our current work are as follows:

Current study is focused on long-term oral administration of NAC in mice under normal diet, through provision of NAC from 18 to 60 weeks of age, provided some protection from beta-cell oxidative stress and damage in response to a HFD challenge (up to 8 weeks of HFD feeding, see attached 2025 experimental timeline) administered at 40 weeks of age. However, the overall effects of long-term NAC treatment in this aging model were ambiguous. Importantly, our current work indicates that antioxidant therapies to support metabolic health during aging need to be carefully considered to ensure that physiological processes are not disrupted.

Schuurman et al. 2022 examines the effects of NAC in HFD-induced obesity and diabetic mouse model. The hypothesis of the study was that the efficiency of NAC treatment will improve metabolic outcomes and beta-cell function by rescuing beta-cell overcompensation while reducing beta-cell stress and PaSC activation induced by HFD. For this work mice received high-fat diet from ages 6-26 weeks and 6-36 weeks of age. NAC was provided as both a prevention (administered one week prior to diet start) and as an intervention (12 weeks after diet start). All measurements for this work were conducted in the HIGH FAT DIET condition, with mice showing severe weight gain, glucose intolerance and insulin resistance. The purpose of this study was to determine not only the optimal timing of NAC treatment in the diabetic condition but also the optimal dosage of NAC (10mM versus 50mM NAC in drinking water). We determined that both timing and dosage of NAC were important for diabetes prevention and intervention, and recommended more research into its use for humans with diabetes.

Our current submitted work examined the effects of long-term NAC in aging mice fed normal diet; it was a time-course study with mice at ages 26, 36, 48 and 60 weeks. The bulk of this study was conducted with NORMAL CHOW DIET fed mice, and examined long-term NAC treatment on the healthy aging condition, not during severe diet-induced obesity, as was done for our previous study. However, we did employ a HFD-stress challenge for a short period of 8-weeks in the present study, beginning at 40 weeks until 48 weeks, to assess whether mice receiving NAC in the normal condition could be protected from a diet-induced stress later in life. We reference our 2022 study and conclude that in parallel with our previous work, NAC seems to be protective in the HFD condition, as we continued to see beneficial effects in the HFD-diet stress model. The major novel findings from this study are that long-term oral administration of NAC has an impact during healthy aging mice. We found that NAC impaired beta cell identity, function and demonstrated through immunofluorescence and western blotting that insulin signaling pathways and SNARE protein expression were downregulated by the long-term use of NAC, in the aging normal diet condition. The conclusions of this study were that although NAC seems to be beneficial during diet-induced obesity, it appears to have a negative impact on beta cell function in the healthy condition and may have induced ‘reductive stress,’ a state of redox imbalance characterized by too few ROS signaling molecules which are essential for physiological cellular functions.

Here we attached a figure, which is accessible in both manuscripts, outlining the study designs and timelines for diet and NAC administrations.

Reviewer 3 Report

The manuscript (antioxidants-3469153) reported the effects of long-term treatment (up to 42 weeks) with N-acetyl-L-cysteine (NAC) in drinking water in male adult black mice on metabolic outcomes, beta-cell function, and pancreatic stellate cells (PaSC) activation. In particular, NAC reduced insulin secretion and insulin signaling in adult mice during aging. In addition, NAC reduced high-fat diet (HFD) challenge-induced beta-cell oxidative stress and preserved nuclear PDX-1 expression. It was concluded that long-term NAC administration in normal diet-fed aging mice may have caused ‘reductive stress’ in islets, resulting in reduced insulin production and secretion.

Overall, the topic is interesting, in particular the finding that long-term NAC treatment may have some adverse effects on islets function in normal diet animals despite its beneficial effects on HFD-challenged mice.

 I have a few comments for the authors:

1.      The mice studied were in the age range of 18-60 weeks. These are healthy adult animals and, in my opinion, are not ‘aged’ mice. I am not sure whether they can be called ‘aging’ mice.

2.      Did NAC also reduce insulin production and secretion in HFD mice? HFD mice had hyperinsulinemia. Was it caused by oxidative stress?

3.      I don’t see Supplementary Table S1 in the supplementary material.

4.      What is the function of PDX-1?

5.      Fig 6F, open triangle should be NACHFD rather than HFDNAC. In the legend, it was 3-5 mice/group. In either Fig 2G or Fig 6F, the significance level was not identified.

6.      At the end of the day, can the authors recommend against chronic NAC in healthy adults and for NAC in unhealthy conditions?

Author Response

RESPONSE TO REVIEWER #1 comments

COMMENT#1      The mice studied were in the age range of 18-60 weeks. These are healthy adult animals and, in my opinion, are not ‘aged’ mice. I am not sure whether they can be called ‘aging’ mice.

We appreciate this comment and want to be clear that we did not refer to these mice as ‘aged’ in the manuscript. We agree that the model we used is not of aged animals, but of aging animals. Throughout the manuscript we referred to the mice as ‘aging’, as we followed them during aging from 18 weeks to endpoints of 28 to 60 weeks of age, to assess biological changes which occur, with or without NAC administration, across the lifespan of the animal.

COMMENT#2      Did NAC also reduce insulin production and secretion in HFD mice? HFD mice had hyperinsulinemia. Was it caused by oxidative stress?

We did not measure insulin production or insulin content within isolated islets of mice undergoing the HFD challenge. HFD-challenge induced beta cell oxidative stress and hyperinsulinemia (plasma insulin level at ~19.6 ng/ml in HFD vs. ~5.2 ng/ml in ND), which was prevented in mice receiving NAC (plasma insulin level at ~6.1 ng/ml). Figure 6F shows that there appears to be significantly higher insulin secretion in HFD mice compared to mice receiving NAC, indicating that NAC prevented islet oxidative-stress could rescue HFD-caused hyperinsulinemia.

COMMENT#3      I don’t see Supplementary Table S1 in the supplementary material.

Supplemental Table 1 has been uploaded to supplementary material.

COMMENT#4      What is the function of PDX-1?

Beta cells maintain a mature beta cell identity through regulation of gene expression using transcription factors such as pancreatic duodenal homeobox (PDX-1). PDX-1 is a transcription factor which plays an essential role in endocrine pancreas development as well as mature beta cell identity and function, beta-cell insulin secretion and survival. We have revised current manuscript with added “When analyzing the transcription factor PDX-1, it involved in beta-cell function and insulin secretion, we found that nuclear PDX-1 localization was significantly lower in 28 and 60-week NAC mice (Figure 4E, 4F)”. This update is reflected in line 233.

COMMENT#5      Fig 6F, open triangle should be NACHFD rather than HFDNAC. In the legend, it was 3-5 mice/group. In either Fig 2G or Fig 6F, the significance level was not identified.

Labeling in Fig6F has been updated. Statistical significance levels have been added to Fig 2G and 6F where applicable.

COMMENT#6      At the end of the day, can the authors recommend against chronic NAC in healthy adults and for NAC in unhealthy conditions?

In the conclusions, we continue to support the use of NAC in the diet-induced obese condition as evidence for use in pathological conditions has been studied in humans as well as rodent models without adverse effects. However, at this time we cannot make clinical recommendations for or against NACs use in healthy adults and urge further research into the impact of antioxidants on beta cell function and signaling pathways.

Round 2

Reviewer 1 Report

The authors did not answer the reviewers' concerns, and did not provide any discussion point, leading to almost no change or revisions in the manuscript. While they provide explanations for their study design, they tend to dismiss potential limitations rather than acknowledge them as areas for further discussion. In some cases, they partially address the questions but avoid directly answering key points, such as the relevance of their model to human metabolic aging or whether additional comparisons could strengthen their findings.

That said, their study remains interesting and valuable, particularly in its exploration of NAC's effects on metabolic health during aging. However, a more open and constructive approach—acknowledging limitations while reinforcing the strengths of their design—would improve the clarity and credibility of their responses, and integrating changes would improve the overall clarity and relevance of the manuscript.

See attached PDF with detailed comments in red text. 

Author Response

RESPONSE TO REVIEWER #3 Round 2 comments [in Red, response in Green]

Comment #1

The authors utilize NAC at 50 mM, which is a relatively high dose for in vivo studies. What are the potential impacts of this dose on the mice? How does this concentration compare to doses used in the literature? Adiscussion on the rationale for selecting this dose and any potential side effects would improve the manuscript. To my knowledge many studies use 1g/L NAC which is equivalent to 6.14 mM NAC, and to my experience this isthe NAC concentration point at which the drinking water gets sour (low pH) and would need to be sweetened (which can vary depending on the alkalinity of the local drinking water). Please clarify, and provide rationaleand explanations. What is the pH of the 50mM NAC water used? Do the authors modify this in any additional way (adding sugar?). If the authors have not modified the drinking water in anyway, please try tasting 50mM NAC water, perfectly safe but very sour.

This methodology was approved in our animal use protocol within guidelines of Canadian Council of Animal Care. The concentrations used in our work compare well with a published study by (Ref: Falach-Malik et al., 2016) where they found ~600-1200mg/kg/day had the best metabolic outcomes. Animals in our study consumed NAC within this range. Our 2022 study also used 10mM drinking water and we found that this concentration was not sufficient to improve metabolic outcomes. We did test the pH in 10mM and 50 mM NAC solutions and found that they are both pH2.3. We did not modify NAC containing drinking water with sugar or other sweetener for this study. The mice drank the NAC-supplemented water, with no adverse health reports or any dehydration noted by our facility animal care staff or researchers due to NAC consumption.

Thank you for this important clarification. Including this explanation in your discussion and study design regarding thechoice of dose is necessary for the clarity of the manuscript. Indeed,is clear to me what you have done, please includethis information in your methods section, as it will be a quality point of reference for future studies.

Thank you for your suggestion, we have included a more detailed description in the methodology (lines 91-95 highlighted). Our previous study details the rationale for selecting a concentration of 50mM NAC as an appropriate dose, and we have included a sentence and citation of this study in the methods that states that the selection of NAC was informed by our previous research.

Comment #2

Age of the mice and its translational relevance: The oldest time point studied is 60 weeks (≈14 months), whichcorresponds to <50 human years (Ref: Flurkey et. al, 2004; Flurkey et. al 2007) or the Mouse in Biomedical Research.

  1. How does this relate to the onset of type 2 diabetes (T2D), insulin resistance, and glucose intolerance in aging humans? Clarifying this would help assess the relevance of the model. The authors should explain how the 60-weeks of age has been chosen.

This study was conducted to determine the effects of long-term NAC during healthy aging mice. Based on ourprevious published work (Oakie et al., 2019 Diabetologia) and in initiating our current aging study, we found that mice after 60 weeks age begin to develop adverse age-related health outcomes such as reduced survival, spontaneous tumour development etc. In order to maintain sufficient number of mice for the current study, we limited the study duration to 60 weeks.

The authors do not fully answer my question. Even during healthy aging, mice can develop insulin resistance and beta-cell dysfunction with aging. Therefore, the provided answer does not clarify the translational relevance of the model tohuman aging and metabolic disorders linked to aging.

The relevance of their model needs to be discussed. Whether the metabolic changes in their model parallel those seen in aging humans at risk for T2D, insulin resistance, and glucose intolerance is important information for the reader and contextualize their findings.

This study was done to assess the impacts of long-term NAC supplementation on physiologically aging mice and its associated changes with glucose tolerance and beta-cell function. While this is not a pathological model of diabetes, there is significant research lacking in the effects of antioxidant supplementation in these non-pathological conditions. We acknowledge that “healthy aging” was a misuse of wording in the previous response. A 60-week time course was analysed because metabolic impairments are minor at this point in the mouse aging process but not completely absent. We specifically did not choose “aged” mice (1.5 years-2 years) to avoid more severe age-related pathologies (such as spontaneous tumour generation) that could confound the purpose of our study, which is to examine the effects of long-term NAC supplementation on regulating glucose tolerance.

Oxidative stress increases with age, as does age-related pathologies such as diabetes, in rodents and humans. Many people take antioxidant supplements to increase their antioxidative capacity as these are readily available to consumers. It has been reported that upwards of 50% of the population consumes dietary supplements (Kantor et al. 2016) and an estimated 54% of dietary antioxidant intake comes from supplements (Chun et al. 2010). There is a lot of research already investigating the use of antioxidants to improve metabolic outcomes in rodents during the diabetic condition and some research on humans with mixed results.

More recent research has shown the importance of ROS signaling in physiological pathways and taking antioxidants prematurely could disrupt redox balance and lead to “reductive stress.” There is very limited research investigating the overuse of antioxidants resulting in reductive stress. Argaev-Frenkel et al (2022, cited Ref. #55) have shown similar impairments to insulin signaling in other tissues following NAC treatment that reflects what we observed with our findings in islets. Our manuscript has been updated to address this and is explained in the response to part b) of this comment.

Reference did not included in the manuscript as listed below:

Kantor ED, Rehm CD, Du M, White E, Giovannucci EL. Trends in Dietary Supplement Use Among US Adults From 1999-2012. JAMA. 2016 Oct 11;316(14):1464-1474. doi: 10.1001/jama.2016.14403. PMID: 27727382; PMCID: PMC5540241.

Chun OK, Floegel A, Chung SJ, Chung CE, Song WO, Koo SI. Estimation of antioxidant intakes from diet and supplements in U.S. adults. J Nutr. 2010 Feb;140(2):317-24. doi: 10.3945/jn.109.114413. Epub 2009 Dec 23. Erratum in: J Nutr. 2010 May;140(5):1062. PMID: 20032488.

  1. Although T2D onset occurs around 45+ years in humans, its incidence increases significantly after 60 years(https://cdc.gov/diabetes/php/data-research/index.html). Therefore, a limitation of the study is that NAC’s effects may be pronounced because metabolic impairments at this age (max 60 weeks) are still relatively mild. The authors need to discuss this point.

  • As mentioned in (a) above, this study was designed to investigate the effects of long-term NAC treatment in healthy aging mice, not in aged diabetic mice. We also examined whether long-term NAC treatment can prevent fat-induced stress leading to the impairment of glucose tolerance and beta-cell function, using a HFD challenge during

As raised in point (a), aging can lead to metabolic impairments, insulin resistance and beta- cell dysfunction. Howdo the authors ensure they are studying “healthy aging”, and not just “aging”

  • Regarding aging related to impaired glucose metabolism, from Figure 1E, you can see modest impairments of glucose tolerance over time, indicated by a trend towards increased AUC of GTT. However, mice receiving NAC appear to have reduced AUC of GTT overtime indicated improved glucose tolerance. We respectfully disagree that the effects of NAC may be more pronounced at this age as numerous studies show significant impacts on GTT and ITT in diabetic and/or severely obese mice which are significantly improved with NAC treatment, as reported in our previous work and by other

Authors mean Figure 2E, not 1E.

Moreover, the authors do not address the core issue raised: the potential limitation that NAC’s effects might bemore pronounced at 60 weeks because metabolic impairments at this age are still relatively mild. While the authors cite prior research showing NAC's effectiveness in more severe models (e.g., diabetic or obese mice), references are not provided. Kindly asking you to provide references.

The authors do not explicitly discuss whether the relatively mild metabolic impairments in their model could amplify NAC’s perceived benefits. This needs to be integrated in the manuscript.

References showing the effectiveness of NAC administration in more severe models (diabetes and obesity) are as follows:

Shen, F.-C., Weng, S.-W., Tsao, C.-F., Lin, H.-Y., Chang, C.-S., Lin, C.-Y., Lian, W.-S., Chuang, J.-H., Lin, T.-K., Liou, C.-W., & Wang, P.-W. (2018). Early intervention of N-acetylcysteine better improves insulin resistance in diet-induced obesity mice. Free Radical Research, 52(11–12), 1296–1310. https://doi.org/10.1080/10715762.2018.1447670

Ma, Y., Gao, M., & Liu, D. (2016). N-acetylcysteine Protects Mice from High Fat Diet-induced Metabolic Disorders. Pharmaceutical Research, 33(8), 2033–2042. https://doi.org/10.1007/s11095-016-1941-1

Cited Ref#16 Schuurman, M., Wallace, M., Sahi, G., Barillaro, M., Zhang, S., Rahman, M., Sawyez, C., Borradaile, N., & Wang, R. (2022). N-acetyl-L-cysteine treatment reduces beta-cell oxidative stress and pancreatic stellate cell activity in a high fat diet-induced diabetic mouse model. Frontiers in Endocrinology, 13(938680). https://doi.org/10.3389/fendo.2022.938680

Cited Ref# 17 Falach-Malik, A., Rozenfeld, H., Chetboun, M., Rozenberg, K., Elyasiyan, U., Sampson, S. R., & Rosenzweig, T. (2016). N-Acetyl-L-Cysteine inhibits the development of glucose intolerance and hepatic steatosis in diabetes-prone mice. American Journal of Translational Research, 8(9), 3744–3756. https://pubmed.ncbi.nlm.nih.gov/27725855/

In our model, we only suggest that NAC is beneficial in the HFD component of this study as the dose of NAC mice received in this study appeared to prevent negative outcomes associated with a HFD-oxidative stress challenge. Although the 60 week mice receiving NAC showed improved glucose tolerance, their islet signaling was impaired and PDX-1, an important transcription factor that regulates beta cell identity, was significantly reduced. Given that ROS is required for beta cell function (Plecita-Hlavata et al., 2020, cited Ref. #49), we expect the extended use of NAC resulted in reductive stress, which led to the impaired signaling observed within the insulin pathway. Further research must be conducted to determine the impacts of impaired signaling long-term and also the general impacts of inhibiting ROS on beta cell function and insulin signaling.

We updated the manuscript to contain a limitation of the translatability to humans and to emphasize how the timeline of our study relates to human age and age-related pathologies. See lines 475-495, highlighted.

Comment #3

Comparison of HFD +/- NAC with ND 48 weeks (no HFD) is lacking. This comparison is necessary to fully apprehend the extent of

i)HFD with aging; ii) NAC rescue effect on HFD with aging.

Although direct comparisons were not made between ND, NDHFD and NACHFD at 48 weeks, all of the ND 48 weeks data is already included in this manuscript in the aging time course branch of the study. This portion of the study wasconducted to determine if NAC could rescue negative metabolic outcomes associated with a HFD-challenge during aging. Comparisons between NDHFD and NACHFD achieves this goal.

The extent of the rescue and NAC effect is difficult to apprehend without knowing the impact of HFD in the first place. If the authors can not provide the data, which is fully understandable because of the extent of the long-term in vivo studies, acknowledging and discussing this limitation is required and would strengthen their manuscript. If the authors cannot provide a discussion for these points then the authors should consider omitting these data from the current manuscript and adjust the overall conclusions accordingly.

First, we would like to emphasize that this HFD challenge study in long-term NAC mice is not to determine how NAC rescued the effect on HFD-induced diabetes. Rather, the administration of NAC at this time-point showed that long-termNAC could protect aging islets against a short-term HFD-induced stress challenge. Adding this part to the study further verified the functional role of NAC on pathological conditions that induce oxidative stress, such as the conditions observed in mice challenged with high-fat diet stress.

We had 4 groups in this study at 48 weeks: ND, NAC, NDHFD and NACHFD. All of the 48-week data from the ND and NAC mice were included in the aging time course branch of the study (Fig. 1ABEG; Fig. 2EF; Fig. 3BD; Fig. 4ACDF). In order to avoid repeating these data again in this manuscript, we had decided to only compare between the ND and NAC cohorts undergoing HFD challenge, as displayed in current figures 6 and 7.

We have also included a summary of findings from the four groups as requested for the reviewer:

Figure 6: Long-term NAC supplementation preserves weight and glucose metabolism in aging mice undergoing HFD challenge. (A) Body weight trends during HFD challenge (n=6 ND and NAC; n=5 NDHFD; n=8 NACHFD mice/group). Body weight (B) (n=6 ND and NAC, n=5 NDHFD; n=8 NACHFD mice/group), fasting blood glucose (C) (n=10 ND; n=6 NAC; n=15 NDHFD; n=12 NACHFD mice/group), and fed plasma insulin (D) (n=9 ND and NAC; n=6 NDHFD; n=8 NACHFD mice/group) after 8 weeks of HFD challenge. (E) IPGTT and (F) AUC (n=7 ND; n=6 NAC; n=6 NDHFD; n=5 NACHFD mice/group), and (G) IPITT and (H) AUC (n=5 ND; n=6 NAC; n=5 NDHFD; n=6 NACHFD mice/group) after 8 weeks HFD challenge. HFD challenge in control diet (NDHFD): closed triangle; Control diet (ND): closed circle; NAC: open circle. HFD challenge in control diet (NDHFD): closed triangle; HFD challenge in NAC treatment group (NACHFD): open triangle. Data are expressed as means ± SEM. *p<0.05, **p<0.01; analyzed using one-way ANOVA followed by Tukey’s Post-Hoc test.

 Fig. 7

Figure 7: Long-term NAC supplementation preserves nuclear PDX-1 expression and reduces beta-cell oxidative stress in aging mice undergoing HFD challenge. Quantification of islet density (A), alpha-cell mass (B), beta-cell mass (C), PDX-1 (D), 8OHdG (E) and (F) αSMA (n=3-4 pancreata/group) ND and NAC mice 8 weeks after HFD challenge. Representative double immunofluorescence images for (G) GLUT2 and (H) Syntaxin 1A labelled in red co-stained with insulin (green); nuclei are stained with DAPI (blue). Scale bars: 50µm. Control diet (ND): closed circle; NAC: open circle. HFD challenge in control diet (NDHFD): closed triangle; HFD challenge in NAC treatment group (NACHFD): open triangle. Data are expressed as means ± SEM. **p<0.01, ***p<0.001; analyzed using one-way ANOVA followed by Tukey’s Post-Hoc test.

Comment #4

In line with previous comment, the rationale for using HFD and studying NAC effect on HFD should be better explained.

We determined with our 2022 published study that NAC is beneficial as a prevention and intervention for improvingnegative metabolic outcomes associated with long-term HFD feeding(>20 weeks), which was conducted in mice from 6 to 28/36 weeks of age. The HFD challenge in the current manuscript was conducted as an extension to our previousstudy, to determine if NAC was effective for prevention of negative metabolic outcomes associated with a HFD challenge during aging.

The authors need to add this clear rationale to their manuscript. Indeed, it will increase the clarity and purpose of their study when framed with a well formulated rationale.

A very similar variation of what was outlined above has already been included in the manuscript introduction rationale (lines 69-74): “Our previous study determined that NAC improved glucose tolerance and insulin sensitivity in high-fat diet (HFD)-induced diabetic mice in a time-and dose-dependent fashion, which was associated with a reduction of beta-cell oxidative stress, intra-islet pancreatic stellate cell (PaSC) activation and islet fibrosis. However, there is a lack of investigation into the effects of long-term NAC on pancreatic beta-cells and intra-islet PaSCs in the context of physiological aging.”

Comment #5

The authors are using pEIF2a as a marker of ER stress (line 262-263). However, EIF2a is a broader marker of the integrated stress response (ISR), and not exclusively of ER stress.

Therefore, the conclusions to this observations (reduced p-EIF2a) should be corrected accordingly (lines 264-267). If the authors want to comment on ER stress they need to evaluate levels of the ER stress branches which include IRE1a -> XBP1 splicing and ATF6 levels respectively.

Thus, If the authors want to check and make conclusions on ER stress status, which is relevant in the context of T2D, other more specific markers must be used.

Thank you for this comment, we have updated to the manuscript to reflect pEIF2a as a marker of the integrated stress response. This update is reflected in line 265, 403-404.

Yes. But please the authors should not speculate about that it could be ER stress, it clearly doesn´t matter for the main message. However, if the authors would like to make a statement about ER stress, then they need to show it with the addition of a couple of western blots on the same lysates of p-PERK t-PERK which would be the ER stress related upstream signal connecting ER stress and the unfolded protein response (UPR) with the integrated stress response(ISR). These are very straightforward blots and clearly within the interest of the authors biology. The authors could possibly also include p-GCN2 and t-GCN2 as the nutrient sensing arm above the ISR, which would probably be aninvestment they would benefit from in both this study and their current in vitro research mentioned in the next point.

To align with the reviewer’s concern, we have removed all statements that suggest p-EIF2a may indicate reduced ER stress. We included a statement saying it would be important in future studies using NAC to investigate any connection between NAC, ER stress and insulin production in beta cells. We appreciate your suggestions to investigate other pathways in the ISR and ER stress and will consider it for future studies. However, it is not feasible to add the present study due to time constraints of this review process. We identified significantly reduced insulin levels in the isolated islets- along with reduced pEIF2a and oxidative stress- and had making a connection with existing literature that has demonstrated that ER stress is required for proper insulin folding in the ER and insulin production (Lui et al, 2018, Haataja et al 2016, Ikezaki et al., 2020, cited Ref. #51-53). It was not intended as a main message of the study.

Comment #6

Pathways investigated in Figure 5 and shown as decreased signaling in NAC-treated mice (ND60) should also beinvestigated in HFD; does the HFD increase these pathways (Fig5A SNARE exocytotic pathway, and 5C Insulinreceptor signaling and AKT/ERK) which were rescued by NAC?

Islet isolations were only conducted on the 60 week old mice in this study. For the HFD- challenge at 48 weeks, we performed in vivo glucose metabolism tests and collected pancreatic tissue for the immunofluorescence staining of mouse islets. We agree it would be interesting to investigate how long-term NAC impacts these pathways during HFD stress, and it is a goal of our current in vitro research. It was also not feasible to preserve the 3 Rs of animal research (as required by our Canadian Council on Animal Care), particularly “reduce,” we would have had to approximatelytriple the number of mice used in the 48 weeks age study in order to achieve the necessary n-value.

That a lot of mice for including one line of observation. But that´s ok, I understand the limitations of any particularstudy design, however I do not believe the 3 Rs of animal research would interfere with well controlled animal experiments.

Thank you for understanding this limitation of our study.

Comment #7

Inconsistency in sample size across figures: the number of mice for a given experimental group varies acrossdifferent panels of the same figure. Example: Figure 1B (body weight, ND60, n=6), Figure 1C & 1D (ND60, n=7), and Figure 1F (ND60, n=8). Why the group for body weight measurement (the less invasive measurement) is smaller than other more invasive measurements? If some analyses were conducted on a subset of animals, this should be clearly stated in the Methods, currently key information missing.

Thank you for drawing this to our attention. The body weight trends and final body weight graphs have been updated in Figure 1A and 1B.

Indeed. I am happy to point out that a lot of data was missing from your figures. It would indeed be a shame from a 3 Rs perspective if many mice from your study were lost due to hasty preparation of figure legends. Indeed, this is why it is so important to explicitly write out the exact n = in your figure legends.

Exact n-values have been added to all figure legends, with the figure legends now stating how many mice or samples were included in each examined group.

Comment #8

It is currently unclear if the authors have conducted one experiment or multiple experiments with long term NAC treatment. The n = needs be defined in every figure and cannot be a range!

Due to the varied n-values across our timepoints, we included individual data points on our graphs to be transparent with readers and so n-values can be observed in each data set.

The authors do not directly address the core concern: whether they conducted one experiment or multiple experimentswith long-term NAC treatment. Please respond and add to your manuscript methods if you have done 1 or multiple in vivo experiments with long term NAC treatment.

How many you have done will not effect the overall assessment of your manuscript. Currently it is not defined and can be perceived as ambiguous.

Additionally, the authors have not updated the n “range”, still in use in the figure legends, PLEASE write the exact n for each figure in the figure legends, one cannot make out the exact n in each figure as the figure quality and size does not allow for it. I do appreciate that this will take 5-10 minutes of the authors time, but please understand that this is not up for debate. If the authors for some reason do not have access to the exact n for their figure panels, they wouldhave to provide an explanation for each instance why they cannot provide the exact n for each respective figure panel.

As stated in the previous response, the exact n-values have now been added to all figure legends.

We are unsure whether there has been a misunderstanding regarding the statement of “one experiment or multiple experiments.”, and we will do our best to address this concern. This study was conducted over a 4-year period. The 28-week and 36 week study were data collected in parallel with our 2022 manuscript investigating NAC in HFD-induced diabetic mouse model. The 48- and 60-week study was conducted over the previous 2 years. We had mice allocated for their pancreata to be paraffin embedded for immunohistological analysis and mice allocated for islet isolation/ex vivoGSIS. Additionally, in accordance with our approved Animal Use Protocol, only 2 metabolic tests can be performed on one mouse (Two of either: IPGTT, IPITT, or in vivo GSIS). Additionally, all n-values represent litter-matched control and NAC treated mice from multiple in vivo experiments. We do not intend to be ambiguous and are open to any recommendation for inclusion of this information in the manuscript text/supplemental materials.

“All n-values represent litter-matched control and NAC treated mice from multiple in vivo experiments” was added to lines 90-91 of the manuscript, highlighted. We hope that this clarifies any confusion with the presentation of these results.

Comment #9

As the authors have conducted a 60 week long study on C57/B6 animals, and stated that “individuals with healthissues were excluded” this need to be better reported especially as it is a longitudinal study and potential adverse effects of NAC treatment or HFD treatment might be selected away from the final analysis. Indeed it is well known that in the pure C57/B6 background animals tend to get spontaneous tumors and lymphomas around 1-year, and it wouldn´t be surprising if HFD and or NAC might impact the outcome of malignancies in such cohorts in light of a plethora of research on NAC/antioxidants and HFD and their impact on tumor progression in the last decade...

“Health issues” in this context referred to issues outside of study conditions, such as animal fighting and dermatitislesions due to over barbering which led to weight changes, and not due to study conditions. We did not investigate the effects of NAC on tumors in this study and cannot comment on that in the context of this study. We did not observenegative health outcomes due to NAC or HFD administration. We would like to emphasize again, any mice that developed health concerns did not develop these concerns as a result of study conditions, and were excluded from the study.

Nevertheless, did any animals enrolled in the current study develop any spontaneous tumors. It is a simple yes or no question. Indeed it is clear based on the content of the manuscript that the authors are not studying conditions such as animal fighting, dermatitis lesions or spontaneous tumors all very normal incidents in C57/B6 mice enrolled in 60week long interventions studies.

To directly answer your yes/no question: no, we did not identify any visual tumours at the time of dissection. That’s not to say age related tumours/lymphomas weren’t present as they are common in aging mice, especially in 60 week intervention studies, but they weren’t apparent to the naked eye when dissecting the pancreata or fat pads of mice. The only mice ever excluded were due to fighting/dermatitis combined with weight loss as the associated weight loss would confound our diet controlled study.

Minor Comment #1

Legend Figure 1; mentioning biweekly weight measurement. Based on graph shown Figure 1A showing measurementevery other week, the authors meant “every other week” which is not the same as two-times every week (4-fold difference)

Biweekly has dual meaning, either every other week or twice weekly. We agree it is an ambiguous terminology. We have updated the figure legend to remove any confusion.

Thank you for fixing this issue, indeed biweekly is at best an ambiguous terminology, but most frequently it is used forevery other week, and in Scientific writing and Scientific communication leaving room for ambiguity is not in line with good practice.

  No further changes were made as this comment was resolved in the previous review phase.

I am looking forward to reading a revised manuscript with above ambiguities resolved.

Reviewer 2 Report

As indicated in the previous version, The proposed correlation between PDX-1 levels and 8OHdG should be explained better. Also, decreased insulin levels in the beta cells from old-treated animals must be better explained (together with the lower fasting serum insulin levels). A decreased GLUT2, Akt, and ERK signaling expression is related to altered insulin secretion in beta cells. Although this correlates with lower circulating insulin and PDX-1 levels, it is considered an unwanted response to treatments, and therefore, it is difficult to explain regarding serum glucose levels in the animals. 

As indicated above further molecular details to support the signaling obtained in beta cells should be included.

Author Response

Response to Reviewer #2 Round 2 Comments

Comment #1 on Methods

Please include the procedure for obtaining isolated islets in the methods. To normalize results among groups, markers of different pancreatic cell types should be used.

The islet isolation protocol has been used routinely in our islet biology lab for the past 30 years with many publications. In order for the reviewer to understand how rodent islet isolation is performed, we have added a brief description in the methods section (lines 151-157): “In brief, 3 ml collagenase V (1 mg/mL, Sigma) was slowly injected into the common bile duct after occlusion of the distal end just proximal to the duodenum. The distended pancreas was excised and the digestion was performed in a water-bath at 37°C for 30 min. The digestion was stopped by ice-cold Hanks’ balanced salt solution containing 10% FBS, and washed thoroughly to remove residual collagenase. Freshly isolated islets from the pancreas of ND and NAC mice at 60 weeks were hand-picked under a dissecting microscope …”

To identify islets following pancreas digestion, we performed manual hand-picking of islets under a dissecting microscope to visually locate small, round, and relatively cell-dense clusters that appear distinct from the surrounding exocrine pancreatic tissue, which is usually more fragmented and less compact. Islets have smooth borders and a light, translucent appearance when viewed under a microscope.

Regarding your question of normalizing results among groups, we have included islet images labeled by glucagon (green) and insulin (red) below. The major endocrine cell type in the islet is beta cells, and because of this, our study did not focus on normalizing markers to other endocrine cells in the islet.

Comment #2 on Results

From the author's statements, and compared with previous article the main difference in experimental setting is the use of a long or short period of HFD and comparison of the results obtained with the Chow diet. These changes regarding the previous manuscript are not very significant, and also the results obtained are at some point difficult to explain (I disagree that a positive result is a decrease in signaling in the beta cells).

Therefore, these results should be justified in more detail and a molecular mechanism should be proposed to support this conclusion.

We did mention our explanation of this in the revised manuscript and research in context during the first round of responses. In lines 69-74 of the manuscript, we do state that: “Our previous study determined that NAC improved glucose tolerance and insulin sensitivity in high-fat diet (HFD)-induced diabetic mice in a time-and dose-dependent fashion, which was associated with a reduction of beta-cell oxidative stress, intra-islet pancreatic stellate cell (PaSC) activation and islet fibrosis [16]. However, there is a lack of investigation into the effects of long-term NAC on pancreatic beta-cells and intra-islet PaSCs in the context of physiological aging.”

There is a key difference in the source of induced beta cell stress in the islets (diet vs aging) and is not due to the fact that one study is short and another is long. Importantly, the addition of HFD-stress challenge in this study was to further verify the functional role of NAC on aging antioxidative stress when combined with pathological oxidative stress condition.

The molecular mechanism is proposed in Figure 8 and the explanation for the proposed mechanism is included in the discussion. We did not propose that decreased insulin signaling was a positive result. In fact, we propose the opposite and suggest that long-term NAC resulted in a state of reductive stress (below physiological ROS level) impairing insulin signaling and production (lines 404-409; 419-424; 428-435). We feel this is interesting research and should be further explored.

Comment #3 on relevant contribution

The article remains very similar to the previous one, and the main point that the authors stress, the response to treatment is different in chow and HF diets, needs more results to support it.

Again, we would like to point out that this research explores the long-term use of NAC in physiologically aging mice. These mice are not experiencing any other pathologies such as diabetes or diet induced obesity, outside of the cohorts used for Figures 6 and 7. It is widely accepted that ROS are essential signaling molecules in various tissues and proper redox balance must be maintained for physiological functions. While beneficial effects of NAC are known to ameliorate pathologies associated with increased oxidative stress, limited research exists on the effects of antioxidants in the non-pathological condition. Therefore, we feel this is important research as the use of antioxidants in humans and animal studies has grown exponentially in recent years with little investigation to their impact after long-term use. We identified that long-term use of NAC in mice is associated with altered insulin signaling in the pancreatic islets and reduced PDX-1, which is considered one of the most important markers of a physiological beta cell. The altered signaling is also different from a classical diabetic islet as there would be elevated stress markers and increased ERK signaling, which was not observed in aging mice receiving NAC. Indeed, the effects of antioxidants on aging require further exploration.

Comment #4 regarding Major comments

As indicated in the previous version, The proposed correlation between PDX-1 levels and 80HdG should be explained better. Also, decreased insulin levels in the beta cells from old-treated animals must be better explained (together with the lower fasting serum insulin levels). A decreased GLUT2, Akt, and ERK signaling expression is related to altered insulin secretion in beta cells. Although this correlates with lower circulating insulin and PDX-1 levels, it is considered an unwanted response to treatments, and therefore, it is difficult to explain regarding serum glucose levels in the animals.

The explanation of insulin signaling pathways are described in Figure 8 and in the discussion (lines 446-460). Here we included a more detailed explanation/correlation of PDX-1 and oxidative stress in lines 399-409. It is interesting that with reduced insulin and PDX-1, NAC treated mice are still able to maintain glucose homeostasis and even improve glucose tolerance. We also showed no changes in insulin sensitivity as measured with IPITT – we did not identify impaired insulin sensitivity as you stated in major comments. However, we suspect there are effects on peripheral tissue such as adipose, muscle and liver. Liver is especially important for its role in glucose metabolism. However, we are an islet biology lab and cannot speculate in detail about how or if NAC may be altering other tissues. This is the first study investigating long-term use of an antioxidant in mice and its specific effect on pancreatic islets/beta cells. There is much to be explored in further studies.

Comment #5 regarding detail comments

As indicated above further molecular details to support the signaling obtained in beta cells should be included.

Molecular mechanisms of our western blot and IHC analysis are summarized and outlined in Figure 8 and in discussion.

Round 3

Reviewer 1 Report

I have no further comments...

My concerns and QnAs were addressed 

Reviewer 2 Report

Although I do not agree with the novelty and significance of the "new data," the manuscript is somewhat improved in the new version.

The manuscript could be acceptable in the present condition.